Herbarium-based studies on taxonomy, biogeography and ecology of Psilochilus (Orchidaceae)

Kolanowska Marta 1 2 martakolanowska@wp.pl
Naczk Aleksandra M. 3
Jaskuła Radomir 4
1 Department of Plant Taxonomy and Nature Conservation/Faculty of Biology, University of Gdańsk , Gdańsk , Poland
2 Department of Biodiversity Research, Global Change Research Institute AS CR , Brno , Czech Republic
3 Department of Molecular Evolution, University of Gdańsk , Gdańsk , Poland
4 Department of Invertebrate Zoology and Hydrobiology, University of Lodz , Łódź , Poland
Roberts David
Electronic publication date: 2016 Nov 8
Publication date: 2016
Volume: 4
Electronic Location ID: e2600
Received 2016 May 20; Accepted 2016 Sep 23
Copyright: © 2016 Kolanowska et al.
Copyright year: 2016
Copyright holder: Kolanowska et al.
License: This is an open access article distributed under the terms of the Creative Commons Attribution License, which permits unrestricted use, distribution, reproduction and adaptation in any medium and for any purpose provided that it is properly attributed. For attribution, the original author(s), title, publication source (PeerJ) and either DOI or URL of the article must be cited.
License URL: https://creativecommons.org/licenses/by/4.0/

Keywords: Ecological niche modeling, Phytogeography, Species richness, New species, Biodiversity, Taxonomy, Psilochilus, Neotropic ecozone

Funding: Faculty of Biology, University of Gdańsk 538-L150-B583-14 European Community Research Infrastructure Action under the FP7 “Capacities” Program GB-TAF-2445 Grantová agentura České republiky (GA ČR) 14-36098G The study was financed by the Faculty of Biology, University of Gdańsk (grant no. 538-L150-B583-14). This research received support from the SYNTHESYS Project (http://www.synthesys.info/) which is financed by European Community Research Infrastructure Action under the FP7 “Capacities” Program (GB-TAF-2445) and from the grant no. 14-36098G of the Grantová agentura České republiky (GA ČR). The funders had no role in study design, data collection and analysis, decision to publish, or preparation of the manuscript.

==============================
Background

Psilochilus is a poorly studied orchid genus distributed from southern Mexico to south-eastern Brazil. A taxonomic revision of this Neotropical endemic based on morphological data is presented.

Material and Methods

Over 170 dried herbarium specimens and flowers preserved in liquid of Psilochilus were analyzed. Morphological variation among examined taxa was described based on multivariate analysis. To evaluate the similarity between niches occupied by various Psilochilus species ecological niche modeling (ENM) was applied. Species richness and the distribution patterns of Psilochilus representatives were analyzed based on squares of 5° latitude and longitude while similarities among floras between biogeographical units were measured using the Bray-Curtis index for presence/absence data.

Results and Discussion

A new species of the P. physurifolius-complex is described based on Central American material. Psilochilus crenatifolius is reduced to the rank of variety as P. macrophyllus var. crenatifolius. A key to 18 accepted Psilochilus species is provided. The illustrations of perianth segments of all recognized taxa are presented. The climatic niches preferred by the particular Psilochilus representatives are well separated based on ecological niche modeling analysis. Their distribution is limited mainly by the isothermality and temperature seasonality. The highest Psilochilus species richness is observed in the North Andean, Panamanian, Brazilian Planalto and Central American biogeographical provinces. A high level of endemism is observed in all those regions as well as Yungas biogeographical province. Most Psilochilus species occur in areas above 800 m of elevation. The populations were most often reported from the tropical rain forest and tropical moist deciduous forest.

Introduction

The orchid genus Psilochilus was described in 1882 (Barbosa Rodrigues, 1882) based on P. modestus. Soon after the description, it was synonymized by Cogniaux (1893) with Pogonia Jussieu (1789), the genus which for a long time was recognized as a large taxon that included rather primitive, tropical, terrestrial species. Those orchids were recognized based on subsimilar tepals, lip ornamented with various forms of calli and having slender gynostemium with an apical anther and soft, mealy pollinia. At present, Pogonia s.s. is considered as a genus embracing just 2–4 species known from north-eastern Asia and eastern North America. Several genera were segregated from Pogonia s.l., e.g. the New World Cleistes Richard ex Lindley (1840), Isotria Rafinesque-Schmaltz (1808) and Triphora Nuttall (1818), as well as Paleotropical Nervilia Commerson ex Gaudichaud-Beaupré (in Freycinet, 1829). These taxa may be distinguished one from another based on structure of pollinia, gynostemium and seeds as well as based on the foliage. Considering those characters, Psilochilus is most similar to Triphora by having entire, apical clinandrium and Eulophia-type seeds. They are, however, easily distinguished on the basis of the anther connection with the gynostemium and the form of the floral bracts. The anther is motile in Psilochilus and rigidly fused with the column apex in Triphora. Floral bracts are dissimilar to the leaves in Psilochilus and foliaceous in the latter genus (Ames, 1922a: Ames, 1922b; Szlachetko & Rutkowski, 2000).

Forty years after the formal description, and based on above mentioned differences, Psilochilus was restored by Oakes Ames (1922a), who transferred to this genus Pogonia macrophylla Lindl. This concept was accepted by subsequent researchers, e.g. Dressler & Dodson (1960), Brieger (1975), Szlachetko (1995) and Pridgeon et al. (2005).

As currently recognized, Psilochilus representatives produce fleshy, villose roots along the creeping rhizome. They are characterized by the leafy stems and plicate leaves. The resupinate, rather inconspicuous flowers are produced in succession and arranged in the terminal raceme. Sepals and petals are free, similar in shape. The lip is more or less clawed, 3-lobed in the apical part and it is ornamented with a single or several calli running along the central part of the disc. The elongate gynostemium is more or less arcuate, slender in the lower half and slightly swollen at the apex.

Pfitzer (1887) classified representatives of Psilochilus (as Pogonia), within Pogonieae, Neottiinae together with i.a. Cleistes and Triphora. Initially Schlechter (1911) also did not recognize Psilochilus as separated genus and he included it (as Pogonia) in Nerviliinae (Polychondreae). After reconsideration (Schlechter, 1926) the author accepted the separateness of Psilochilus which he transferred to Vanillieae together with Triphora and Monophyllorchis Schltr. and these three genera were placed by Dressler & Dodson (1960) within Pogoniinae. Ames (1922b) was the first to suggest the distinctiveness of Triphoreae from Pogoniinae, and the subsequent chromosome number-based studies of Baldwin & Speese (1957) confirmed this assumption. The results of this research was applied by Brieger (1975) who modified Schlechter’s classification system. The author transferred Nervilia and Triphora to Nerviliinae and retained Psilochilus and Monophyllorchis in Pogoniinae. Dressler (1979) placed Psilochilus within Triphoreae based on the lack of a clearly incumbent anther, sinuous epidermal cell walls, and an abcission layer between the ovary and perianth. This tribal classification of the genus is currently widely accepted (Dressler, 1993; Szlachetko, 1995; Rothacker, 2007) and it was confirmed in the molecular studies (Chase et al., 2003; Chase et al., 2015). Szlachetko (1995) included Psilochilus in Vanilloideae, but the widely accepted position of the genus is subfamily Epidendroideae (e.g. Dressler, 1993; Chase et al., 2015).

Psilochilus is rather difficult study object. Collections deposited in the herbaria are relatively poor which is a consequence of Psilochilus growth type, its flowers colouration, and its habitat preferences. The plants usually grow in the very thick litter layer in shady forests, and their flowers are rather inconspicuous, usually greenish; consequently, they are difficult to find during field studies. Moreover, dried flowers of Psilochilus are very fragile and they are often damaged in dried specimens. Insufficient material which is often poorly preserved may partially explain the problems encountered when defining Psilochilus relationships with other genera and thereby with placing it in the existing taxonomic systems. For year-even the species delimitation within the genus was based mainly on the length of the leaf petiole and the lip claw size, but the recent studies revealed great variation of the lip shape within Psilochilus and resulted in description of new taxa (Kolanowska, 2013a; Kolanowska, 2013b; Kolanowska, 2014a; Kolanowska, 2014b; Kolanowska, 2015; Kolanowska & Szlachetko, 2012; Kolanowska & Szlachetko, 2013; Kolanowska et al., 2015). Up to date 18 Psilochilus species have been described. An additional specific name, “guatemalensis” was used by Schlechter in 1926, but this taxon has been never formally described and there is no information about representative specimens of this orchid. The reported geographical range of the genus extends from southern Mexico and the Caribbean to Brazil in the south (Rothacker, 2007).

The most important contribution to the knowledge on the genus was made by Rothacker (2007) who made phylogenetic study based on plastid trnL-F intergenic region. Unfortunately he was unable to obtain molecular data for Psilochilus carinatus and P. dusenianus. Moreover, the lists of representative specimens of each taxon provided by this author included numerous individuals lacking flowers hereby their identification is doubtful and may result in incorrect conclusions.

The objective of this contribution is to provide a comprehensive synopsis of Psilochilus that includes morphological characteristics of each species representative, illustration of the perianth segments, notes on their taxonomic affinities and information about habitat and distribution. The general variation pattern of morphological characters among the recognized taxa was analysed. The additional questions raised in this study concerned variation in climatic niches preferences between Psilochilus representatives and differences in species composition within biogeographical units.

Material and Methods

Morphological study

Over 170 dried herbarium specimens (Fig. 1, Annex 1) and flowers preserved in liquid deposited or borrowed from herbaria AMES, COL, F, K, MEDEL, MO, NY, R, UGDA, US, and W were examined according to the standard procedures. Herbaria acronyms are cited in this paper according to Thiers (2015). Each studied specimen was photographed and the data from the label were taken. The leaf form (blade shape and size, petiole length), floral bracts and ovaries were studied first. The perianth segments were studied after softening flowers in boiling water and they were then examined under a stereoscopic microscope. Plants from 97 collections were not possible to identify on the species level due to the incomplete material – usually lack of flowers or damaged lip.

Figure 1 Localities of specimens examined during the study which could be placed on map: specimens with complete flowers possible to identify (gray spots), specimens incomplete, not possible to identify on the species level (white spots). Map generated in ArcGis 9.2.

The electronic version of this article in Portable Document Format (PDF) will represent a published work according to the International Code of Nomenclature for algae, fungi, and plants (ICN), and hence the new names contained in the electronic version are effectively published under that Code from the electronic edition alone. In addition, new names contained in this work which have been issued with identifiers by IPNI will eventually be made available to the Global Names Index. The IPNI LSIDs can be resolved and the associated information viewed through any standard web browser by appending the LSID contained in this publication to the prefix “http://ipni.org/.” The online version of this work is archived and available from the following digital repositories: PeerJ, PubMed Central, and CLOCKSS.

Morphometry

In the morphometric study, selected specimens were described by 19 floral (1–19) and six vegetative characters (20–25; Table 1). A total of 57 specimens for which it was possible to obtain measurements of all investigated traits were included in the analysis. The collected data was used to describe patterns of morphological variation among the studied taxa. All morphometric data were standardized prior to analysis and subsequently subjected to principal components analysis Principal component analysis (PCA; Sneath & Sokal, 1987). The analyses in this part of the work were performed with the use of the three databases: 1) for the floral characters only; 2) for the combined vegetative and floral characters; and 3) for the characters indicated by PCA analysis with the greatest contributions. To determine the morphological characters that differentiate the studied taxa the most, the discriminant analysis was applied, in order to reduce the data set by selecting only the characters that showed strongest discrimination. Statistical computations were performed with the program package: STATISTICA v. 10 (StatSoft Inc., 2010) and PAST v. 2.14 (Hammer, Harper & Ryan, 2001).

Table 1 List of morphological characters studied among the recognized Psilochilus taxa.

Code	Morphological character	
1	Lip–length	
2	Lip–width	
3	Lip–claw length	
4	Lip–lateral lobes length	
5	Lip–lateral lobes width	
6	Lip–middle lobe length	
7	Lip–middle lobe width	
8	Lip–isthmus length	
9	Lip–width of the middle lobe base	
10	Lip–presence of the elevated lamellae on the disc	
11	Lip–length between lip isthmus and lateral lobe apex	
12	Lip–length between claw base and apex of the lip middle lobe	
13	Lip–length between claw base and lip isthmus	
14	Dorsal sepal–length	
15	Dorsal sepal–width	
16	Petal–length	
17	Petal–width	
18	Lateral sepal–length	
19	Lateral sepal–width	
20	Leaf–max length of leaf petiole	
21	Leaf–length of the biggest stem leaf	
22	Leaf–width of the biggest stem leaf	
23	Leaf–length of the leaf subtending the inflorescence	
24	Leaf–width of the leaf subtending the inflorescence	
25	Inflorescence length	

Climatic niche similarity

To evaluate the similarity between niches occupied by various Psilochilus species ecological niche modeling (ENM) was applied. A database of localities was prepared based on the examination of herbarium specimens. For the analysis, only those localities which could be precisely placed on the map were used. The georeferencing process followed Hijmans et al. (1999). The geographic coordinates provided on the herbarium sheet labels were verified. If there was no information about the latitude and longitude on the herbarium sheet label, we followed the description of the collection site and assigned coordinates as precisely as possible to this location. Google Earth (Google Inc.) was used to validate all gathered information. The species for which exclusively one location could be placed on map were excluded from the analysis.

The maximum entropy method, as implemented in Maxent version 3.3.2 software, was used to create models of the suitable niche distribution (Phillips, Dudík & Schapire, 2004; Phillips, Anderson & Schapire, 2006). As Maxent is relatively robust against collinear variables, all 19 available climatic factors, in 2.5 arc-minutes (±21.62 km2 at the equator) as developed by Hijmans et al. (2005) and provided by WorldClim (http://www.worldclim.org/), were used together with the altitudinal data (Table 2). To assess the level of specificity of the analysis, the maximum iterations of the optimization algorithm were established as 10,000 and the convergence threshold as 0.00001. The “random seed” option was used for selecting training points. The run was performed with 1,000 bootstrap replications and the default logistic model was used. The differences between the niches occupied by the studied species were evaluated using the niche overlap test as available in ENMTools: Schoener’s D (D) and I statistic (I) as available in ENMTools v. 1.3 (Schoener, 1968; Warren, Glor & Turelli, 2008; Warren, Glor & Turelli, 2010).

Table 2 Variables used in the ENM analysis.

Code	Description	
bio1	Annual mean temperature	
bio2	Mean diurnal range = mean of monthly (max temp–min temp)	
bio3	Isothermality (bio2/bio7) * 100	
bio4	Temperature seasonality (standard deviation * 100)	
bio5	Max temperature of the warmest month	
bio6	Min temperature of the coldest month	
bio7	Temperature annual range (bio5–bio6)	
bio8	Mean temperature of the wettest quarter	
bio9	Mean temperature of the driest quarter	
bio10	Mean temperature of the warmest quarter	
bio11	Mean temperature of the coldest quarter	
bio12	Annual precipitation	
bio13	Precipitation of the wettest month	
bio14	Precipitation of the driest month	
bio15	Precipitation seasonality (coefficient of variation)	
bio16	Precipitation of the wettest quarter	
bio17	Precipitation of the driest quarter	
bio18	Precipitation of the warmest quarter	
bio19	Precipitation of the coldest quarter	
alt	Altitude	

Biogeography, species diversity and distribution

The study site includes the part of Neotropical realm here defined as tropical Americas. The area is delimited in latitude by the Tropic of Cancer in the Northern Hemisphere and the Tropic of Capricorn in the Southern Hemisphere. According to Udvardy (1975) tropical Americas belong to 37 biogeographical provinces. Based on the FRA 2000 Report (2001) the area is divided into 10 ecological/vegetation zones.

Species richness and the distribution patterns of Psilochilus representatives were analyzed based on squares of 5° latitude and longitude. In each square the number of all species recorded was summarized. Similarities among floras between biogeographical units (Fig. 2) were measured using the Bray-Curtis index for presence/absence data with PRIMER 6 software (Clarke & Gorley, 2006). Jaccard’s (1902) index was used to present the degree of dissimilarity separately between biogeographic regions distinguished by Udvardy (1975): R = 100c/a+b−c, where: a, number of species in the richest flora; b, number of species in the poorest flora, c, number of species common to both floras.

Figure 2 Biogeographical provinces of tropical Americas.

Numbers are given only for units in which Psilochilus species occurrence has been noted: (A) Mardean-Cordilleran, (B) Central American, (C) Panamanian, (D) Cuban, (E) Greater Antillean, (F) Venezuelan Dry Forest, (G) Lesser Antilean, (H) Northern Andean, (I) Llanos, (J) Yungas, (K) Amazonian, (L) Brazilian Plantanto, (M) Serra do mar (classification follows Udvardy, 1975).

The altitudinal distribution of particular taxa were measured using STATISTICA v. 10 (StatSoft Inc., 2010). All records were located using Google Earth software.

Results

Genus composition

As revealed in this study Psilochilus includes 18 species, however the additional taxa may be recognized in the future when additional material will be available. The key to identification of all known genus representatives is provided below. The morphological characteristic of each species and illustration of its perianth segments are presented. During examination of the herbarium material several specimens which based on their morphology could not be classified within any currently known Psilochilus representative were found. The unidentified collections are here shortly characterized in the chapter “Incertæ sedis.”

Taxonomic treatment

Psilochilus Barbosa Rodrigues (1882: 272). Type:—Psilochilus modestus Barbosa Rodrigues (1882: 272).

Terrestrial herbs with decumbent rhizomes rooted at nodes. Roots fleshy, villose. Stem slender, erect, remotely several-leaved. Leaves fleshy, sessile or shortly petiolate, basally expanding into sheaths. Flowers small, resupinate, sessile or subpedunculate in a terminal raceme. Sepals and petals free. Lip free, clawed, usually with two thickenings at the claw base, distally 3-lobed. Gynostemium elongate, more or less arcuate, slender and delicate in the lower half, slightly swollen at the apex. Column foot rudimentary. Anther sessile, erect, oblong-ellipsoid, motile. Pollinia narrowly oblong, powdery. Stigma ventral, slightly concave. Rostellum formed from the apical margin of the middle stigma lobe, erect, truncate, slightly thickened.

Key to identification of Psilochilus species

Lip middle lobe distinctly clawed, claw and apical part of the middle lobe subequal in length2

–Lip middle lobe sessile or subsessile with a claw much shorter than apical part of the middle lobe3

Tepals not falcate, lip middle lobe broadly ovate to suborbicular, lateral lobes not reaching apical part of middle lobe clawP. dusenianus (1)

–Tepals falcate, lip middle lobe elliptic, lateral lobes extending to apical part of middle lobe clawP. hatschbachi (2)

Apex of the lip middle lobe retuse4

–Apex of the lip middle lobe rounded, obtuse to subacute5

Lip middle lobe almost twice longer than wide, narrowly ellipticP. sanderianus (3)

–Lip middle lobe equally long and wide, suborbicularP. steyermarkii (4)

Lip lateral lobes exceeding to 2/3 of the middle lobe lengthP. dressleri (5)

–Lip lateral lobes usually not exceeding middle lobe length6

Isthmus between lip lobes inconspicuous7

–Isthmus between lip lobes distinct8

Lip middle lobe up to 4 mm wide, subquadrate-rounded to triangularP. carinatus (6)

–Lip middle lobe 6.5–10 mm wide, subrhombic to transversely elliptic P. vallecaucanus (7)

Lip claw inconspicous9

–Lip claw prominent11

Lip lateral lobes relatively smallP. szlachetkoanus (8)

–Lip lateral lobes prominent10

Leaves shortly petiolateP. alicjae (9)

–Leaves sessile or subsessileP. macrophyllus (10)

All leaves small, blade up to 2.8 cm longP. minutifolius (11)

–Leaf blade at least 4 cm long12

Middle leaves sessile or subsessile13

–Middle leaves shortly petiolate14

Lip middle lobe narrowly elliptic P. tuerckheimii (12)

–Lip middle lobe suborbicularP. antioquiensis (13)

Lip middle lobe about equally long and wide15

–Lip middle lobe longer than wide17

Leaves acuminateP. physurifolius (14)

–Leaves obtuse to acute16

Leaf apex acute, lip 14–15 mm long, middle lobe up to 5 mm long and wideP. maderoi (15)

–Leaf apex obtuse, lip 17–21 mm long, middle lobe 4–7.5 × 5–8 mmP. modestus (16)

Lip lateral lobes subacuteP. mollis (17)

–Lip lateral lobes rounded to subobtuse at the apicesP. panamensis (18)

1. Psilochilus dusenianus Kraenzlin ex Garay & Dunsterville (in Dunsterville & Garay, 1965: 274).

Type: BRAZIL. Paraná: Monte Alegre, 850 m, P. Dusén 9022 (holotype S!–only photo seen). Figure 3.

Figure 3 Psilochilus dusenianus.

(A) Dorsal sepal. (B) Petal. (C) Lateral sepal. (D–E) Lip. Scale bars = 10 mm. (A–D) Redrawn by N. Olędrzyńska from Dunsterville & Garay (1965), (E) Drawn from Liesner & Carnevali 22537 (MO).

Plants up to 30 cm tall. Leaves 4–5, shortly petiolate; blade up to 10 × 5.5 cm, ovate to ovate-lanceolate, acute; petiole 1–1.5 cm long. Inflorescence about 3 cm long, few-flowered. Floral bracts up to 11 mm long. Pedicellate ovary up to 20 mm long. Flowers pale greenish-yellow. Dorsal sepal 28–30 × 5 mm, oblong-elliptic, acute, 3-veined. Lateral sepals 25–28 × 4–5 mm, obliquely oblong-elliptic, acute, 3-veined, middle vein thickened. Petals 24–27 × 4–5 mm, obliquely oblong-elliptic, acute, 3-veined. Lip 18–21 mm long, 7–9 mm wide across lateral lobes; claw 3–5 mm long; lateral lobes exceeding up to 2/3 of the middle lobe claw, obliquely ovate, rounded or obtuse at the apex; middle lobe clawed, claw 3–7 mm long, above 4–5 × 3–5 mm, broadly ovate to suborbicular, obtuse to rounded; disc with three central thickened veins or lamellae. Gynostemium up to 17 mm long, arcuate in the apical third or fourth.

Distribution, habitat and ecology: This species was described based on Brazilian material collected in forest at the altitude of 850 m (Fig. 4A). It also occurs in Venezuela where it grows in semi-opened forest and at the altitude of about 1,400 m. Flowering in Venezuela occurs in October. The highest niche overlap was observed between P. dusenianus and P. carinatus (D = 0.0131, I = 0.0749), but morphologically the two species are easily distinguished. Unfortunately the lack of sufficient data on the distribution of P. hatschbachi does not allow to compare their climatic preferences.

Figure 4 Distribution of Psilochius species.

(A) P. dusenianus (triangle), P. hatschbachi (star), P. steyermarkii (diamond), P. dressleri (square), and P. carinatus (circle), (B) P. vallecaucanus (square), P. szlachetkoanus (star), P. alicjae (diamond), P. macrophyllus (circle), P. minutifolius (triangle), and P. tuerckheimii (cross), (C) P. antioquiensis (triangle), P. maderoi (square), P. physurifolius (circle), P. modestus (star), P. mollis (diamond), and P. panamensis (cross). Map generated in ArcGis 9.2.

Taxonomic notes: From similar P. hatschbachi this species is distinguishable based on oblique, but not falcate tepals and middle lobe form which is broadly ovate to suborbicular (elliptic in P. hatschbachi).

Additional specimen examined:

VENEZUELA. Territorio Federal Amazonas: Dept. Rio Negro, Cerro Aracamuni summit, 1,400 m, 27 October 1987, R. Liesner & G. Carnevali 22537 (MO!).

2. Psilochilus hatschbachi Kolanowska (2014a: 83).

Type: BRAZIL. Paraná: Curitiba, Quatro Barras. Morro Mãe Catira, 110 m, 12 January 1967, G. Hatschbach 15684 (holotype: F!, isotypes F!, US!). Figure 5.

Figure 5 Psilochilus hatschbachi.

(A) Dorsal sepal. (B) Petal. (C) Lateral sepal. (D–E) Lip. Scale bars = 10 mm. (A–D) Drawn from Hatschbach 15684 (F), (E) Drawn from Hatschbach 15684 (US).

Plant up to 32 cm tall. Leaves 3–5, shortly petiolate; blade up to 8 × 2.3 cm, narrowly ovate, obtuse; petiole up to 1.2 cm long. Inflorescence about 6 cm long, several-flowered. Floral bracts up to 7 mm long. Pedicellate ovary 17–20 mm long. Flower cream-lavender, lip flushed dark red. Dorsal sepal 21 × 3 mm, oblong-lanceolate, obtuse to subacute, 5-veined. Lateral sepals 22 × 3.6 mm, falcate, linear-lanceolate, subacute, 3-veined. Petals 20 × 3 mm, falcate, linear-lanceolate, obtuse, 5-veined. Lip 17 mm long, 7 mm wide across the lateral lobes; claw 3–5 mm long; lateral lobes extending up to the apical part of the middle lobe claw, obliquely ovate, acuminate to obtuse; middle lobe clawed, claw about 2.5 mm long, above 3–4.5 × 2–3 mm wide, elliptic, obtuse, apical margin incurved; disc ornamented with 5 thickened veins and three delicate lamellae extending from the basal third up to the middle lobe center. Gynostemium about 15 mm long.

Distribution, habitat and ecology: So far this species is known from a single collection made in lowland Brazil (Fig. 4A), at the altitude of about 110 m. Flowering occurs in January. This species was not included in niche overlap test.

Taxonomic notes: The only species which may be confused with P. hatschbachi is P. dusenianus described above. Unlike the latter species the lip middle lobe of P. hatschbachi is elliptic and its lateral sepals and petals are falcate. In the first description of P. hatschbachi presence of lamellae on the lip disc was given as an additional difference between the two species, however examination of additional herbarium material revealed that this character is not constant within populations and sometimes the lamellae are reduced to the prominent thickenings on the lip.

3. Psilochilus sanderianus Kolanowska (2014a: 82).

Type: BRAZIL. Sine loc. Imported by F. Sander & Co. (holotype K!). Figure 6.

Figure 6 Psilochilus sanderianus.

(A) Dorsal sepal. (B) Petal. (C) Lateral sepal. (D) Lip. Scale bars = 10 mm. Drawn from Sander & Co. s.n. (K).

Lower part of the stem absent in holotype. Upper leaves sessile to subsessile; blade 5.6–6.7 cm × 2.1–2.5 cm, narrowly ovate; petiole about 0.5 cm long. Inflorescence about 2.3 cm long, several-flowered. Floral bracts about 12 mm long. Pedicellate ovary about 16 mm long. Dorsal sepal 26 × 4 mm, oblong-lanceolate, obtuse, 3-veined. Lateral sepals 20.2 × 3.7 mm, oblong-lanceolate, obtuse, falcate, 3-veined. Petals 21 × 3.8 mm, oblong-lanceolate, subacute, falcate, 1-veined. Lip 21 mm long, 7 mm wide across lateral lobes; claw 5.6 mm long; lateral lobes extending up to about 1/3 of the middle lobe length, obliquely ovate, obtuse; middle lobe 8 × 4.6 mm, sessile, narrowly elliptic, apex slightly retuse; disc with three somewhat thickened central veins running along the whole lip length. Gynostemium is not seen.

Distribution, habitat and ecology: So far, this species is known from a single collection. The plant was imported by F. Sander and collaborators from Brazil and unfortunately nothing is known on the habitat preferences of P. sanderianus. This species was not included in niche overlap test.

Taxonomic notes: Based on floral characters this species resembles P. modestus, but it is easily distinguished from this orchid by the narrowly elliptic lip middle lobe which is retuse at the apex (vs suborbicular).

4. Psilochilus steyermarkii Kolanowska (2015: 32).

Type: VENEZUELA. Territorio Federal Amazonas: Cerro Duida, southeastern-facing sandstone bluffs near Caño Negro (tributary of Caño Iguapo), 1,095–1,520 m, 26 August 1944. J. Steyermark 58052 (holotype F!). Figure 7.

Figure 7 Psilochilus steyermarkii.

(A) Dorsal sepal. (B) Petal. (C) Lateral sepal. (D) Lip. Scale bars = 10 mm. Drawn from Steyermark 58052 (F).

Plant about 20 cm tall, erect. Leaves 4, shortly petiolate; blade up to 7.5 × 2.7 cm, broadly lanceolate to narrowly ovate, acute; petiole about 0.5–1 cm long. Inflorescence about 3.5 cm long, few-flowered. Floral bracts up to 12 mm long. Pedicellate ovary about 20 mm long. Flowers greenish. Dorsal sepal 18–20 × 2.8–3.2 mm, narrowly oblong-lanceolate or linear-elliptic, subobtuse to acute, 3-veined. Lateral sepals 17.5–20 mm long, about 2.5–3 mm wide, narrowly oblong-elliptic, acute to subobtuse, 3- or 5-veined. Petals 16–17 × 2.1–2.6 mm, narrowly oblong-lanceolate or linear-oblanceolate, subacute to obtuse, 3-veined. Lip 18.5 mm long, 5 mm wide across lateral lobes; claw inconspicuous, less than 1.5 mm long; lateral lobes extending not more than up to the half of the middle lobe, triangular, subacute to obtuse, diverging from the middle lobe; middle lobe 4–4.5 × 4.2–4.5 mm, suborbicular, slightly retuse at the apex, margins minutely crenate; disc with three slightly thickened veins running along the lip center. Gynostemium about 14–16 mm long, arcuate in the apical third.

Distribution, habitat and ecology: This species was described based on material collected in Venezuelan Amazonian region where it was found at the altitude of 1,095–1,520 m. Flowering in Venezuela occurs in August. The additional specimen collected in Haiti was found in NY herbarium. Plants from this collection are characterized by shorter leaf petioles than those observed in the type specimen. Figure 4A.

Taxonomic notes: In its habit P. steyermarkii resembles P. physurifolius, but may be easily distinguished from this species by the triangular lip lateral lobes (vs lateral lobes obliquely ovate) and prominent lip middle lobe (vs inconspicuous) which is slightly retuse at the apex (vs rounded). Moreover, in P. steyermarkii the apices of lip lateral lobes extend to the basal 1/3 of the middle lobe (vs lateral lobes not reaching the base of the middle lobe). In the lip form P. steyermarkii resembles P. modestus which, however is characterized by the lip about twice longer than wide (up to 13 × 6 mm) with a short, but distinct claw and prominent triangular lip lateral lobes.

Additional specimens examined:

HAITI. Montagnes de la Hotte, 24 August 1927, W. J. Eyerdam 348 (NY!).

VENEZUELA. Territorio Federal Amazonas: Cerro Duida, southeastern-facing sandstone bluffs near Caño Negro (tributary of Caño Iguapo), 1,095–1,520 m, 26 August 1944, J. Steyermark 58052 (F!).

5. Psilochilus dressleri Kolanowska (2014b: 55).

Type: PANAMA. Prov. Darién: Ridge north of Cerro Pirre, 1,050–1,200 m, 12 July 1977, R. L. Dressler 5663 (holotype FLAS!). Figure 8.

Figure 8 Psilochilus dressleri.

(A) Dorsal sepal. (B) Petal. (C) Lateral sepal. (D) Lip. Scale bars = 5 mm. Drawn from Dressler, 5663 (FLAS).

Plant about 36 cm tall. Leaves about 6, sessile to subsessile; blade 6–7 × 2.9–3.4 cm, ovate, subacute; petiole less than 0.5 cm long. Inflorescence about 2.5 cm long, few-flowered. Floral bracts up to 10 mm long. Pedicellate ovary 16 mm long. Sepals and petals pale green, lip cream with purple mark. Dorsal sepal 19 × 2 mm, concave, oblong-lanceolate, obtuse to subobtuse, 3-veined. Lateral sepals 18 × 2 mm, falcate, linear, subacute, 3-veined. Petals 17 × 1.5 mm, slightly falcate, linear, subobtuse to acute, 3-veined. Lip 14 mm long, 6 mm wide across the lateral lobes; claw about 6 mm long; lateral lobes large, extending to 2/3 of the middle lobe, internal parts overlapping the middle lobe, obliquely ovate-falcate, obtuse; middle lobe 4 mm long and about the same wide, suborbicular, obtuse, margins entire; disc with 3 thickened veins. Gynostemium is about 16 mm long.

Distribution, habitat and ecology: Known so far exclusively from the Darién Gap (Fig. 4A), where it was found growing in wet forest at the altitude of about 1,050–1,200 m. Flowering occurs in July.

Taxonomic notes: This species resembles Psilochilus macrophyllus (Lindl.) Ames with relation to their subsessile leaves but it differs by the prominent, large lateral lobes of the lip that extends to two-third of the middle lobe and relatively small middle lobe which is almost twice shorter than lateral lobes and suborbicular in outline, with entire margins. In P. macrophyllus the obliquely oblong-ovate lip lateral lobes extends usually to about half of the middle lobe length and the lip middle lobe is not distinctly smaller than the lateral lobes.

6. Psilochilus carinatus Garay (1978: 2).

Type: COLOMBIA. Magdalena: Sierra Nevada de Santa Marta, W. Purdie s.n. (holotype K!). Figure 9.

Figure 9 Psilochilus carinatus.

(A) Dorsal sepal. (B) Petal. (C) Lateral sepal. (D–E) Lip. Scale bars = 10 mm. (A–D) Redrawn by N. Olędrzyńska from Garay’s drawing of F. Holton s.n. (K), (E) Drawn from Lectae 2908 (US).

Plants up to 30 cm tall. Leaves about 4, shortly petiolate; blade up to 9.5 × 4 cm, ovate-elliptic, acute or subacuminate; petiole up to 1 cm long. Inflorescence about 6 cm long, few-flowered. Floral up to 15 mm long, bracts ovate-lanceolate. Flowers green, lip tinged with purple or lilac with median part greenish and lateral lobes white. Pedicellate ovary up to 15 mm long. Dorsal sepal 20–23 × 3–3.5 mm, linear-oblong, acute, 1- or 3-veined. Lateral sepals 18–23 × 2.5–3 mm, falcate-linear, acute, 1-veined. Petals 17–20 × 3–4 mm, falcate, oblanceolate to linear-elliptic, acute, 3-or 5-veined. Lip 16–21 mm long, 7 mm long across lateral lobes; claw 5–7 mm long; lateral lobes reaching just a base of the middle lobe, obliquely triangular-ovate, obtuse; middle lobe 3–4.5 × 4 mm, subquadrate-rounded to triangular, apex truncate to acute, margins erose-dentate; disc with three thickened, central veins. Gynostemium is 13–17.5 mm long, slightly arcuate in the apical third.

Distribution, habitat and ecology: This species occurs in Colombia, Ecuador and Bolivia (Fig. 4A). It was also reported from Costa Rica and Panama (Rothacker, 2007), but no specimens from this region were found in herbarium material. Ecuadorian population of P. carinatus was found at the altitude of about 1,525 m, growing in premontane wet forest. Flowering in this country occurs in February. No data on habitat preferences of this species from other Andean countries are available. The highest niche overlap was observed between this species and P. macrophyllus var. brenesii (D = 0.5583; I = 0.8017).

Taxonomic notes: In the clawed lip and petiolate leaves this species resembles P. maderoi. Both species, however, may be distinguished based on the form of the lip middle lobe. The middle lobe of P. carinatus is subquadrate-rounded to triangular while in P. maderoi it is suborbicular.

Additional specimens examined:

BOLIVIA. Sine loc. M. B. Lectae 2908 (US!).

COLOMBIA. Cundinamarca: Fusagasuga, F. Holton s.n. (K!).

ECUADOR. Napo: Archidona, Reserva de Biósfera Sumaco. Vertiente norte del Volcán Sumaco. Comunidad Pacto Sumaco, 1,525 m, 18 February 2003, W. Farfán 426 (MO!).

7. Psilochilus vallecaucanus Kolanowska & Szlachetko (2012: 352).

Type: COLOMBIA. Valle del Cauca: Mun. Cali, KM 18 road Cali-Buenaventura, ca 2,020 m, 16 December 2010, M. Kolanowska 201 (holotype COL!, isotype COL!; UGDA!–drawing). Figure 10.

Figure 10 Psilochilus vallecaucanus.

(A) Dorsal sepal. (B) Petal. (C) Lateral sepal. (D) Lip. Scale bar = 10 mm. Drawn by N. Olędrzyńska from Kolanowska 201 (COL).

Plant up to about 50 cm tall. Leaves up to 5, sessile; blade 4.5–7 × 2–3 cm, ovate, acute, the blade of the lowest leaves extremely reduced. Inflorescence about 5 cm long, few-flowered. Floral bracts 1–1.5 cm long. Pedicellate ovary about 19–20 mm long. Sepals greenish-white, petals white, lip white with the violet or pink margins, middle veins of the tepals darker and thicker than lateral ones. Dorsal sepal 22–23 × 3–3.5 mm, narrowly elliptic, slightly concave, acute, 5-veined. Lateral sepals 19 × 4 mm, falcate, slightly concave, obtuse, 5-veined. Petals 18–19 × 3–4 mm, slightly falcate, obtuse to shortly acuminate, margins of the apical part slightly irregular and undulate, 3-veined. Lip 15–19 mm long, 8–10 mm wide across the lateral lobes; claw 4–9 mm long; lateral lobes 5–10 × 1.25–2.25 mm, falcate, triangular, rounded at the apex; middle lobe 4–5 × 6.5–10 mm, subrhombic to transversely elliptic, apex with a minute, obtuse appendix, margins of the middle lobe undulate; disc with 3- or 5-thickened central veins. Gynostemium about 18 mm long, slightly arched.

Distribution, habitat and ecology: This species in known only from the Western Cordillera of Colombia (Fig. 4B) where it grows in the cloud forests at the altitude of about 1,875–2,020 m. Two small populations of 1–3 individuals were found during fieldwork growing in the thick litter layer. Flowers of P. vallecaucanus seem to be cleistogamous. The most similar niches are occupied by P. macrophyllus var. brenesii (D = 0.1198; I = 0.3371).

Taxonomic notes: This species is vegetatively similar to P. macrophyllus, from which it differs by having a long-clawed lip with obscure, obtuse lateral lobes. P. vallecaucanus is similar in the shape and size of the lip to the specimen found in Ecuador and marked by Rothacker (2007) as LJ5414 (QCA). However, according to the description of this specimen provided by the author, it differs from P. vallecaucanus by having petiolate leaves and papillate lip.

Additional specimens examined:

COLOMBIA. Valle del Cauca: Mun. Yotoco, Hacienda Hato Viejo, 3°49′48″N 76°26′02″W, 1,875 m, 17 February 2010, M. Kolanowska & O. Pérez s.n. (COL!); Finca Zingara. Km 4 via a Dapa, corregimiento de la Elvira, cordillera Occidental, 3°30′N, 76°34′W, bosque de niebla, 1,900 m, 12 February 1994, J. Giraldo-Gensini 168 (MO!).

8. Psilochilus szlachetkoanus Kolanowska, sp. nov.

Species distinguished by the petiolate leaves, short lip claw and minute lip lateral lobes.

Type: MEXICO. Chiapas: Finca El Suspiro, near Berriozambal, September 1957, R. L. Dressler 2257 (holotype US!). Figure 11.

Figure 11 Psilochilus szlachetkoanus.

(A) Dorsal sepal. (B) Petal. (C) Lateral sepal. (D) Lip. (E) Gynostemium. Scale bars = 5 mm. Drawn from Dressler, 2257 (US).

Plant 20–25 cm tall. Leaves 3–5, shortly petiolate; blade 5–7 × 3.5–4 cm, ovate, subobtuse; petiole 1–1.8 cm long, the upper leaves subsessile. Floral bracts 0.8–1.4 cm long. Inflorescence about 3–4 cm long, few-flowered. Flowers pink according to the note on herbarium label. Dorsal sepal 21–22 × 2–2.5 mm, oblong-elliptic to linear-oblanceolate, subacute, 3-veined. Lateral sepals 19–19.5 × 3–4 mm, narrowly elliptic-oblong, slightly oblique, obtuse to subacute, 3-veined. Petals 17–20 × 2.5–3.2 mm; oblong-elliptic above narrow base, obtuse, 3-veined. Lip 15–17 mm long, 5–6 mm wide across lateral lobes; claw about 1–2.5 mm long; lateral lobes minute, not reaching 1/4 of the middle lobe length, obliquely ovate, rounded at the apex; middle lobe 5–5.2 × 4–5.7 mm, ovate, apex subobtuse, margins slightly crenulate; disc with three central thickened veins. Gynostemium 14–15 mm long, slightly arcuate in the upper third.

Distribution, habitat and ecology: So far this species is known from Mexico and Costa Rica (Fig. 4B). In Mexico it was found flowering in September growing in cloud forest. The highest niche overlap was observed between the new species and P. macrophyllus var. macrophyllus (D = 0.1297; I = 0.3315). The same statistic calculated for P. physurifolius was relatively low (D = 0.0135; I = 0.0614).

Etymology: Dedicated to D. L. Szlachetko, Polish orchidologist.

Taxonomic notes: The new species resembles representatives of P. physurifolius-complex by the distinctly petiolate leaves, but the lip shape of the new entity allows to easily distinguish it from other species of Psilochilus. The obscure lip lateral lobes are observed in P. vallecaucanus and P. steyermarkii, but none of those two species has large, ovate middle lobe with obtuse apex. From P. vallecaucanus the new species is additionally distinguished by the petiolate leaves.

Additional specimen examined:

COSTA RICA. Alajuela: La Palma de San Ramon, A. M. Brenes 1113 (NY!).

Additional information: Foldats (1969) in Flora of Venezuela provided drawing of species identified as P. macrophylllus, but this illustration is consistent with the characteristic of the new species described above (shortly petiolate leaves, large lip middle lobe).

9. Psilochilus alicjae Kolanowska (2014a: 83).

Type: BRAZIL. Paraná: Serra do Mar, Ypiranga in silvia primara ad terram, 15 January 1914, P. Dusén 14461 (holotype NY!, isotype K!). Figure 12.

Figure 12 Psilochilus alicjae.

(A) Dorsal sepal. (B) Petal. (C) Lateral sepal. (D–E) Lip. Scale bars = 5 mm. (A–D) Drawn from Dusén 14461 (K), (E) Drawn from Farfán 523 (MO).

Plant up to about 26 cm tall. Leaves 3–4, petiolate; blade up to 9 × 3.8 cm, narrowly ovate to ovate, obtuse; petiole up to 1.6 cm long. Inflorescence up to 7 cm long, several-flowered. Floral bracts 8–12 mm long. Pedicellate ovary up to 17 mm long. Flowers greenish with white lip with lilac apex. Dorsal sepal 16 × 3 mm, linear-lanceolate, obtuse, 3-veined, concave in the natural position. Lateral sepals 24 × 5.6 mm, falcate, oblong-elliptic, acute, 5-veined. Petals 21 × 4 mm, falcate, narrowly elliptic, 5-veined. Lip 18 mm long, 7 mm wide across the lateral lobes; claw about 2.5 mm long; lateral lobes extending to about half of the middle lobe, obliquely ovate, rounded at the apex; middle lobe shortly clawed, about 8.5 mm long in total, 4.8–5 mm wide, ovate to rhombic in outline, apex subacute, margin minutely crenate, disc with five slightly thickened middle veins. Gynostemium about 15 mm long.

Distribution, habitat and ecology: Known from Brazil and Ecuador (Fig. 18) where it was found growing in the premontane wet forest at the altitude of about 1,700 m. Flowering occurs in January and March.

Taxonomic notes: Vegetatively this species resembles P. maderoi and P. modestus, but the lip form allows to easily distinguish those taxa. The lip claw of P. maderoi is distinctly longer than in P. alicjae. In P. modestus the lip middle lobe is sessile, suborbicular and obtuse at the apex.

Additional specimen examined:

ECUADOR. Napo: Archidona, Parque Nacional Sumaco-Galeras. Cumbre de la Cordillera de Galeras, 1,690 m, 11 March 2003, W. Farfán 523 (MO!).

10. Psilochilus macrophyllus (Lindl.) Ames (1922a: 45). ≡ Pogonia macrophylla Lindley (1858: 335).

Lectotype (designated by Kolanowska, Szlachetko & Kras, 2014): CUBA. Sine loc., 1856–1857, C. Wright 615 (K-L!). Figures 13–15.

Figure 13 Psilochilus macrophyllus var. macrophyllus.

(A) Dorsal sepal. (B) Petal. (C) Lateral sepal. (D–F) Lip. Scale bars = 5 mm. (A–D) Redrawn by N. Olędrzyńska from Hamer (1984), (E) Drawn from Bucher 6670 (NY), (F) Drawn from Ekman 9000 (US).

Figure 14 Psilochilus macrophyllus var. brenesii.

(A) Dorsal sepal. (B) Petal. (C) Lateral sepal. (D–F) Lip. Scale bars = 5 mm. (A–D) Drawn from Brenes 247(1434) (F), (E) Drawn from Jimenez 1240 (US), (F) Drawn from Ackerman, 2619 (US).

Figure 15 Psilochilus macrophyllus var. crenatifolius.

(A) Dorsal sepal. (B) Petal. (C) Lateral sepal. (D) Lip. Scale bars = 5 mm. Drawn by S. Nowak from Valeur 723 (K).

Plants up to about 40 cm tall. Leaves 2–6, sessile or subsessile; blade 3.5–6 × 2–3 cm, ovate, acute. Inflorescence 2–14 cm long, several- to many-flowered. Floral bracts up to 20 mm long. Pedicellate ovary about 26 mm long. Flowers creamy-white, pale yellow, greenish-yellow with lip suffused purple, or white with lip with a central green band. Dorsal sepal 11–23 × 2–3 mm, linear-oblong, acute to acuminate, 3-veined. Lateral sepals 11–20 × 2–5.5 mm, linear-oblong to linear-elliptic, acute to acuminate, 3- or 5-veined. Petals 10–18 × 2–3.5 mm, linear to linear-oblanceolate, obtuse, 3- or 5-veined. Lip 12–15 mm long, 5.5–7.5 mm wide across lateral lobes, claw 2.5–5 mm long; lateral lobes obliquely oblong-ovate, usually obtuse; middle lobe 4–6 × 3.5–5 mm, suborbicular to ovate or elliptic, margin crisped, erose; disc with 2-, 3- or 5 thickened veins. Gynostemium 13–18 mm long, slightly arched.

Previous recognition: For years all specimens of Psilochilus with sessile leaves were identified as P. macrophyllus so the geographical range of this species was very wide, extending from Mexico and the Caribbean to Peru and Guyana (Foldats, 1969; McLeish, Pearce & Adams, 1995; Schultes, 1960; Ackerman, 1995; Schweinfurth, 1970; Carnevali & Ramírez-Morillo, 2003; Rothacker, 2007). Not many authors provided drawings of the specimens examined, but at least some of the reports are doubtful. The specimen from Trinidad and Tobago (Schultes, 1960) corresponds to P. physurifolius as well as flower presented by Ames (1922a). As mentioned before also illustration provided by Foldats (1969) does not match the description of P. macrophyllus. The actual geographical range of the species seems to be more limited (Fig. 18).

Taxonomic notes: This species is characterized by the sessile or subsessile leaves, shortly clawed lip and prominent lip lateral lobes. The other species with sessile or subsessile leaves may be easily distinguished from P. macrophyllus based in lip shape. The differences are discussed and illustrated in this paper.

Morphological studies on the variation of the flower morphology within P. macrophyllus revealed that at least two forms of the lip shape are produced among populations. The differences between sessile-leaved Psilochilus was also noticed by Dressler (2003) who, however, did not decide to formally separate the two recognized groups. Based on conducted studies and considering the variation within populations two conclusions may be drawn. Previously described P. crenatifolius seems to fall into lip shape variation of P. macrophyllus hereby it is reasonable to reduce it to the rank of variety of P. macrophyllus. A total of three varieties within P. macrophyllus can be distinguished base on the leaf margin form and shape of the lip middle lobe. Since P. macrophyllus var. crenatifolius is known from a single locality it was not included in niche ovarlap test. This statistic calculated for other two taxa indicate that the climatic conditions preferred by these orchids are almost identical conforming the close relation between them.

1. P. macrophyllus (Lindl.) Ames var. macrophyllus

This variety is characterized by the entire leaf margins and ovate to suborbicular lip middle. Figure 13.

Distribution, habitat and ecology: The confirmed records of this variety come from the Greater Antilles (Jamaica, Cuba, Haiti, Dominican Republic), the Windward Islands (Dominica) and South America (Colombia and Brazil) (Fig. 4B). Populations were found growing at the altitudes of 150–1,850 m. Flowering occurs in February, March and April, June, August, and October. It was found growing in the deep shade or mor humus in rain forest as well as in woods on slopes.

Additional specimens examined:

BRAZIL. Paraná: Marumbí, 800 m, 13 February 1904, P. Dusén s.n. (R!).

COLOMBIA. Valle del Cauca: San Antonio, TV tower, D. L. Szlachetko 9158 (UGDA!).

CUBA. Loma Gardero S. Maeitro, 1 August 1935, J. T. Roig, G. C. Bucher 6670 (NY!), Oriente, crest of Sierra Maestra between Pico Turquino and La Bayamesa, 1,350 m, 27–28 October 1941, C. V. Morton, J. Acuna 3526 (US!).

DOMINICA. St. Paul, trail leading to Morne Trois Pitons, 2,500 ft, 14 June 1967, D. C. Wasshausen, E. S. Ayensu 389 (US!), Syndicate, St. Peter. Near Picard gorge. March–April 1996, C. Whitefoord 7358 (BM!).

DOMINICAN REPUBLIC. “Las Abejas,” wet wooded valley about 10 miles W from Aceitillar. Limestone and baucite. Decumbent or erect, foliage purple and green, 1,200–1,300 m, 24 February 1969, A. H. Liogier 14199 (NY!, US!).

HAITI. Massif de la Hotte, western group, ca. 900 m, 27 August 1927, E. L. Ekman 9000 (US!).

JAMAICA. St. Andrew: Mt. Horeb peak. Upper montane rainforest, 450 ft, 1 February 1977, A. C. Podzorski JA12 (K!).

2. P. macrophyllus (Lindl.) Ames var. brenesii Kolanowska, var. nov.

This variety is characterized by the entire leaf margins and narrowly elliptic lip middle lobe.

Type: COSTA RICA. La Palma, 1,125 m, 29 September 1925, A. M. Brenes 247(1434) (holotype F!). Figure 14.

Distribution, habitat and ecology: This variety was not found in South America. The records come from the Greater Antilles (Jamaica, Dominican Republic, Puerto Rico), the Windward Islands (Martinique), Guatemala, Costa Rica and Panama. Populations were found growing at the altitudes of 830–1,500 m (Fig. 4B). Flowering occurs in June, August and September. Populations were found growing in wet forest understory and in damp, shady forested land.

Additional specimens examined:

DOMINICAN REPUBLIC. Santiago: Terrestre, en lugar húmedo y somrio sobrío la cima del Pico Igua, 960 m, 15 August 1946, J. de J.S. Jimenez 1240 (US!).

GUATEMALA. Huehuetenango: Vicinity of Maxbal, about 17 miles north of Barillas, Sierra de los Cuchumatanes, 1,500 m, 15–16 July 1942, J. A. Steyermark 48897 (F!).

JAMAICA. Vinegar Hill, 4,000–5,000 ft, 25 June 1896, W. Harris 6252 (K!).

MARTINIQUE. Camp Colson, September 1899, A. Duss 4484 (NY!, US!).

PANAMA. Coclé: El Valle de Anton, crest of Cerro Pajito, 1,100 m, 28 September 1946, P. H. Allen 3756 (MO!).

PUERTO RICO. Maricao: Maricao Forest Reserve, western end of Las Tetas de Cerro Gordo above Rd 120, 830 m, 30 June 1989, J. D. Ackerman 2619 (MO!, US!).

3. P. macrophyllus (Lindl.) Ames var. crenatifolius (Kolan.) Kolanowska, comb. et stat. nov.

Basionym: Psilochilus crenatifolius Kolanowska (2013b: 832).

Type: DOMINICAN REPUBLIC. Santiago: District of San José de Las Matas, Arroyo Jiconié, 750 m, 8 October 1930, E. J. Valeur 723 (holotype K!, isotypes MO!, NY, US!). Figure 15.

This variety is characterized by the crenate leaf margins and ovate lip middle lobe.

Distribution, habitat and ecology: So far known only from the Dominican Republic where it was found growing among rocks at the altitude of about 750 m. Flowering occurs in October (Fig. 4B).

11. Psilochilus minutifolius Kolanowska (2015: 36).

Type: PANAMA. Coclé: Near summit of Cerro Gaital, N of El Valle de Antón, 9 July 1982. R. L. Dressler 6073 (holotype FLAS!). Figure 16.

Figure 16 Psilochilus minutifolius.

(A) Dorsal sepal. (B) Petal. (C) Lateral sepal. (D) Lip. Scale bars = 5 mm. Drawn from Dressler, 6073 (FLAS).

Plant less than 12 cm tall. Leaves 5, shortly petiolate; blade up to 2.8 × 2.1 cm, broadly ovate, obtuse; petiole 0.6–0.9 cm long. Inflorescence up to 2.5 cm long, few-flowered. Floral bract up to 11 mm long. Pedicellate ovary 16 mm long. Flowers green. Dorsal sepal 18 × 2 mm, linear-lanceolate, subobtuse, 1-veined. Lateral sepal 16–17 × 4 mm, falcate, narrowly elliptic, subacute, 1-veined. Petals 16 × 2.5 mm, falcate, oblong-lanceolate, subacute, 1-veined. Lip 16 mm long, 6 mm wide across lateral lobes; claw 4.2 mm long; lateral lobes not reaching half of the middle lobe length, obliquely ovate, obtuse; middle lobe 4.9 × 5.2 mm, sessile, subrhombic, subacute; disc with three slightly thickened veins running along the lip center. Gynostemium about 16 mm long.

Distribution, habitat and ecology: This species is known exclusively from the slopes of Panamanian Cordillera Central (Fig. 4B). Flowering occurs in July. The collector’s note on the herbarium label suggests that the flowers of this species are apparently cleistogamous.

Taxonomic notes: The small, broadly ovate leaf blades was not observed in any other Psilochilus representatives. From P. physurifolius this species may be easily distinguished by the subrhombic, subacute lip middle lobe (vs middle lobe suborbicular to broadly ovate, rounded or obtuse). In the lip form P. minutifolius resembles P. macrophyllus from which it clearly differs by the petiolate leaves as well as in the shape of the leaf blade (broadly ovate vs narrowly elliptic to lanceolate-ovate). Unlike P. minutifolius the lip middle lobe of P. modestus is suborbicular.

12. Psilochilus tuerckheimii Kolanowska & Szlachetko (2013: 310).

Type: GUATEMALA. 5,000 ft, January 1878, H. von Türckheim 52 (holotype W!). Figure 17.

Figure 17 Psilochilus tuerckheimii.

(A) Dorsal sepal. (B) Petal. (C) Lateral sepal. (D) Lip. Scale bars = 5 mm. Drawn by S. Nowak from von Türckheim 52 (Holotype: W).

Plant up to about 35 cm tall. Leaves 2–4, subsessile; blade 4–6 × 2–3 cm, narrowly ovate, subobtuse; petiole less than 0.8 cm long. Inflorescence 5–7 cm long, several-flowered. Floral bracts about 5 mm long. Pedicellate ovary 15–20 mm long. Dorsal sepal 18–20 × 3.8–4 mm, somewhat concave, elliptic to oblanceolate, apex obtuse, 1- or 3-veined. Lateral sepals 16–18 × 2.7–4 mm, oblong-oblanceolate, somewhat falcate, subobtuse, 1- or 3-veined. Petals 15.5–17 × 3–3.2 mm, narrowly elliptic, somewhat falcate, subacute, 5-veined. Lip 14–16 × 6 mm; claw 2.5–3.5 mm long; lateral lobes not reaching 1/3 of the middle lobe length, obliquely elliptic, apices rounded at the apex, distant from the middle lobe, curved, directed inwards; middle lobe 3.5–5 × 1.5–2 mm, sessile, narrowly elliptic, obtuse at the apex; disc with one or three median thickened vein(s). Gynostemium 13–14 mm long.

Distribution, habitat and ecology: So far known only from Guatemala (Fig. 4B) where it grows at the altitude of 1,500–1,600 m. Flowering occurs in January and December.

Taxonomic notes: This species belongs to the P. macrophyllus complex characterized by the relatively short lip claw and sessile or subsessile leaves. However, the lip form allows to easily distinguish P. tuerckheimii from other genus representatives. The apices lateral lobes of this species are falcate, rounded, distant from the middle lobe and directed inwards. In P. macrophyllus the lip lateral lobes run close to the middle lobe.

Additional specimen examined:

GUATEMALA. Alta Verapaz: 1,600 m, December 1907, H. von Türckheim II1998 (US!).

13. Psilochilus antioquiensis Kolanowska (2013a: 116).

Type: COLOMBIA. Antioquia: Mun. Jardin, Microcuenca El Clavel, Reserva Natural Cuchilla Jardín Támesis, 5°36′20″N 75°46′30″W, 2,000–2,400 m, 18 May 2006, J. A. Pérez Zabala et al. 2619 (holotype MEDEL!, UGDA!–drawing). Figure 18.

Figure 18 Psilochilus antioquiensis.

(A) Dorsal sepal. (B) Petal. (C) Lateral sepal. (D) Lip. Scale bars = 5 mm. Drawn by N. Olędrzyńska from Pérez Zabala et al. 2619 (MEDEL).

Plant over 20 cm tall (the holotype is damaged in the lower part). Leaves few, just two are present in the holotype, sessile; blade 6–9 × 2–3 cm, ovate, acute. Floral bracts 1–1.6 cm long. Inflorescence about 15 cm long, few-flowered. Flowers greenish, lip with violet margins. Dorsal sepal 20.5–22 × 4.5–4.75 mm, narrowly elliptic above narrow base, obtuse to subacute, 3-veined. Lateral sepals 19–19.5 × 3–4 mm, narrowly elliptic-oblong, slightly oblique, obtuse to subacute, 3-veined. Petals 16–17 × 2.5–3 mm; linear, obtuse, 1-veined. Lip 14.5–15.5 mm long, 6 mm wide across lateral lobes; claw about 3–3.5 mm long; lateral lobes not exceeding the base of the middle lobe, triangular-falcate, apices obtuse to subacute; middle lobe 3–3.5 mm long and the same wide, suborbicular, apex obtuse to subacute, margins slightly crenulate; apices of the lateral lobes distant from the base of the middle lobe; disc with three central thickened veins. Gynostemium 12–13 mm long.

Distribution, habitat and ecology: So far this species is known only from the type locality (Fig. 4C). The population was found growing terrestrially in the litter of the montane forest at about 2,000–2,400 m alt. Flowering occurs in May.

Taxonomic notes: Vegetative characters suggest the relation of this species with P. macrophyllus from which it differs by the short lip lateral lobes, reaching only the basal part of the middle lobe and which apices are distant from the base of the middle lobe. The floral characters make P. antioquiensis similar to P. physurifolius from which it differs by the sessile leaves and a short lip claw.

14. Psilochilus physurifolius (Rchb.f.) Løjtnant (1977: 168) ≡ Pogonia physuraefolia Reichenbach (1859: 324).

Lectotype (here designated): GUYANA. R. Schomurgk s.n. (K!). Figure 19.

Figure 19 Psilochilus physurifolius.

(A) Drawing of the flower of P. physurifolius type specimen placed on the holotype’s sheet. Redrawn by N. Olędrzyńska. (B) Drawing of the lip presented by Rothacker (2007). Redrawn by N. Olędrzyńska. (C) Lip, drawn from Polanco 4018 (PMA).

Plants up to 50 cm tall. Leaves 3–4, shortly petiolate; blade 5–9 × 2.5–3 cm, narrowly elliptic to elliptic-lanceolate, acuminate; petiole 0.5–1 cm long. Inflorescence 2–7 cm long, several-flowered. Flowers pale green, lip with purple suffusion. Floral bracts about 7–8 mm long. Pedicellate ovary about 1.5 cm long. Flower pale green, petals greenish-white, lip whitish with rose middle lobe or dull red markings. Dorsal sepal 19–22 × 3–4 mm, oblong-oblanceolate to oblong-elliptic, subacute to obtuse, 3-veined. Lateral sepals 17–21 × 3–3.5 mm, falcate, linear-lanceolate to linear-elliptic, subacute, 1- or 3-veined. Petals 18–19 × 2–2.5 mm, falcate, linear-oblong, subobtuse, 5-veined. Lip 15.5–16.5 mm long, 5 mm wide across lateral lobes; claw 4–8 mm long; lateral lobes not exceeding middle part of the middle lobe, obliquely ovate, acute to obtuse; middle lobe 4–5.4 × 3.2–4.8 mm, suborbicular to broadly ovate, rounded to obtuse, sometimes shortly apiculate; disc with three thickened veins running along the center. Gynostemium 16–18 mm long, slightly arcuate.

Distribution, habitat and ecology: This species was described based on material collected in Guyana. In this study the occurrence of P. physurifolius was noticed in Grenada, Costa Rica and Panama. It was also reported from Guyana and Venezuela (Rothacker, 2007), but this information was not confirmed. The illustration provided by Schultes (1960) of the specimen from Trinidad and Tobago also corresponds to P. physurifolius. Populations of this species from Central America (Fig. 4C) were found growing at the altitudes of 1,200–1,400 m. Flowering in this region occurs in August and November. The highest, but still very low, niche overlap was observed between P. physurifolius and P. carinatus (D = 0.0155, I = 0.87927).

Taxonomic notes: The lip shape of this species varies among populations––in some the lateral lobes are short, reaching about 1/3 of the middle lobe length, other are characterized by the lateral lobes extending almost to the middle part of the middle lobe. All representatives of P. physurifolius have lip with conspicuous claw and distinctly petiolate leaves. The distinction of this species from similar P. modestus based on the ovate to elliptic, apically obtuse or shortly acute leaves observed in this species. In P. physurifolius the leaves are acuminate, more elongate than in P. modestus.

Additional specimens examined:

COSTA RICA. Cartago: Forest on ridge between Quebrada Casa Blanca and road to Transito, Tapantí, 1,400 m, 10 August 1984, M. H. Grayum & B. Jacobs 3722 (MO!).

GRENADA. Sine loc. 1890–1891, R. V. Sherring s.n. (K!).

PANAMA. Darién: Ridgetop area north of Cerro Pirre, between Cerro Pirre top and Rancho Plastico, 1,200–1,400 m, 14 November 1977, Folsom et al. 6300 (MO!), Panama: Camino entre la cima maxima y la segunda cima. Hierba terrestre de 30 cm, J. Polanco 4018 (PMA!).

15. Psilochilus maderoi (Schltr.) Schlechter (1926: 180) ≡ Pogonia maderoi Schlechter (1920: 41).

Type: COLOMBIA. Cauca: Sine loc. 1,500 m, M. Madero s.n. (B†). Figure 20.

Figure 20 Psilochilus maderoi.

Original Schlechter’s (1929) drawing of Pogonia maderoi.

Plants up to 30 cm tall, erect. Leaves shortly petiolate; blade 6–7 × 2–2.5 cm, elliptic, acute; petiole up to 1 cm long. Inflorescence about 8 cm long, several- to many-flowered. Floral bracts 5 mm long. Pedicellate ovary up to 15 mm long. Flowers greenish, lip suffused with purple. Dorsal sepal 17 × 3–4 mm, narrowly oblanceolate to linear-oblanceolate, acuminate, 1-veined. Petals 15 × 4–4.5 mm, falcately narrowly oblanceolate, acute, 5- or 7-veined. Lateral sepals 17 × 4 mm, falcately linear-ligulate to oblong-ligulate, acute, 1-veined. Lip 14–15 mm long, 7 mm wide across lateral lobes; claw 5–7 mm long; lateral lobes not reaching half of the middle lobe length, elliptic-ovate, obtuse to rounded; middle lobe up to 5 mm long and wide, suborbicular, obtuse, undulate along margins; disc with elevated keel running along the lip center. Gynostemium 12–14 mm long.

Distribution, habitat and ecology: This species is Colombian endemic. It grows terrestrially in premontane forest at the elevation of 1,500–2,000 m (Fig. 4C).

Taxonomic notes: Based on Schlechter’s description and drawing, both P. modestus. P. maderoi and P. modestus may be easily distinguished by the lip claw length (claw very long in P. maderoi).

Additional specimen examined:

COLOMBIA. Cauca: Rio Vangolis on the highlands from Popayán, 1,700–2,000 m, F. C. Lehmann 10038 (K!).

Additional information: Garay’s illustration of perianth segments of the specimen collected by Lehmann 10038 shows flower with relatively short claw, however it was not made based on mature flower but the flower bud. It is considered here as representative of P. maderoi based on the short leaf petiole.

16. Psilochilus modestus Barbosa Rodrigues (1882: 273) ≡ Pogonia modesta (Barb. Rodr.) Cogniaux (1893: 133).

Type:—illustration of Rodrigues (1877). Figure 21.

Figure 21 Psilochilus modestus.

(A) Dorsal sepal. (B) Petal. (C) Lateral sepal. (D) Lip. Scale bar = 5 mm. Redrawn by N. Olędrzyńska from the original Barbosa Rodrigues’s illustration.

Plant up to about 40 cm tall. Leaves about 4, shortly petiolate; blade 8 × 3.8 cm, ovate to elliptic, obtuse; petiole up to 1.5 cm long. Inflorescence 4–9 cm long, 4–15-flowered. Floral bracts up to 12 mm long. Pedicellate ovary up to 16 mm long. Flowers greenish. Dorsal sepal 18–30 × 3–4 mm, narrowly elliptic-oblong, obtuse, 5-veined. Petals 17–24 × 2.6–4.8 mm, falcate, narrowly-elliptic to oblanceolate, obtuse, 3- or 5-veined. Lateral sepals 18–28 × 3–5.4 mm, falcate, linear-elliptic, acute, 5-veined. Lip 17–21 mm long, 10 mm wide across lateral lobes; claw 4.5–7 mm long; lateral lobes extending to the base of middle lobe, obliquely ovate, obtuse, slightly crenate along margins; middle lobe 4–7.5 × 5–8 mm, suborbicular, obtuse, margins creante; disc with three keels or thickened veins. Gynostemium 15–20 mm long, arched.

Distribution, habitat and ecology: This species was found in Colombia, Ecuador and Brazil (Fig. 4C). It was reported also from Venezuela, Nicaragua (Rothacker, 2007) and Costa Rica (Dodson, 1992), but those records were not confirmed. In Ecuador it was reported growing in cloud forest. Populations of P. modestus were found at the altitudes up to 1,960 m. Flowering occurs in January, February and July. The highest niche overlap was observed between P. modestus and P. mollis (D = 0.0251, I = 0.1090), while the score of this statistic for two most similar species, P. physurifolius was extremely low (D = 0.0001; I = 0.0009).

Taxonomic notes: Psilochilus modestus resembles P. physurifolius and P. macrophyllus. From the latter species it may be distinguished based on the petiolate leaves and distinctly clawed lip––those characters are not observed in P. macrophyllus. Psilochilus modestus is easily distinguishable from P. physurifolius based on leaves form which are acuminate in the latter orchid.

Additional specimens examined:

BRAZIL. Rio de Janeiro: Matto, vicinity of Macieiras, Mt. Itatiaya, Esração biologica, 1,960 m, 9 January 1929, L. B. Smith 1770 (US!), São Paulo: Angatuba, Fazenda do Serviço Florestal, 23 February 1966, M. Emmerich 2841, R. Dressler (K!), Paraná: Brejatuba, Mun. Guaratuba. 3–5 m, 5 February 1987, J. M. Silva s.n. (US!).

COLOMBIA. Cundinamarca: Fusagasuga, F. Holton s.n. (NY).

ECUADOR. Morona-Santiago: Cordillera del Cóndor, Cuangos, 20 km east of Gualaquiza, near disputed Peru-Ecuador border. Cloud forest, 1,500 m, 19 July 1993, A. Gentry 80242 (MO!).

Additional information: Specimen collected by J. M. Silva s.n. deposited in US herbarium differs somewhat from typical representatives of P. modestus by having purple flowers (according to the note provided on the herbarium label) and abnormally long lip claw which constitutes almost 1/3 of the total lip length.

17. Psilochilus mollis Garay (1978: 47).

Type: ECUADOR. Morona-Santiago: Río Chihuasi, 25 km SE of Logroño, Cordillera de Cutucú, 800–1,000 m, 16 January 1976, M. Madison & F. R. Coleman 2564 (holotype SEL, isotypes AMES, MO!). Figure 22.

Figure 22 Psilochilus mollis.

(A) Dorsal sepal. (B) Petal. (C) Lateral sepal. (D–E) Lip. Scale bars = 5 mm. (A–D) Drawn from Madison & Coleman 2564 (MO). (E) Redrawn from Garay’s, 1402 illustration deposited in K.

Plant up to 60 cm tall. Leaves 2–4, shortly petiolate; blade 5.5–9 × 3–5 cm, narrowly ovate to ovate-lanceolate, acuminate; petiole 0.5–1.5 cm long. Inflorescence 4–5 cm long, several-flowered. Floral bracts up to 9 mm long. Pedicellate ovary up to 21 mm long. Sepals and petals light green, sepals sometimes suffused with purple, lip white with two longitudinal purple stripes, with or without yellow in the throat with or without violet spots. Dorsal sepal 18–25 × 2.5–3.5 mm, linear to lanceolate, subacute to acute, 1- or 3-veined. Lateral sepals 16–22 × 2.8–3 mm, linear-oblanceolate, acute, 3-veined. Petals 12–22 × 2–3.5 mm, oblanceolate, obtuse to acute, 3- or 5-veined. Lip 20–25 mm long, 6–10 mm wide across lateral lobes; claw up to 10 mm long; lateral lobes extending behind 1/3 of the midlobe, falcate to triangular, subacute; middle lobe 4.5–5.8 × 3.5–5.2 mm, suborbicular to subrhombic, obtuse, margins crenulate-erose; disc with 3–5 thickened veins running along the center. Gynostemium 12–20 mm long.

Distribution, habitat and ecology: This species is found in Ecuador (Fig. 4C) and it was reported also from Peru (Rothacker, 2007). In Ecuador it grows at the altitudes of 800–1,450 m. Flowering occurs in January and February. The highest niche overlap was observed between P. mollis and P. carinatus (D = 0.4474; I = 0.7382).

Taxonomic notes: The examination of the isotype of Psilochilus mollis revealed some inconsistency with the Garay’s drawing deposited in K and the one presented in species description (Garay, 1978). The lip claw of the specimen deposited in MO is very long, constituting about half of the lip length (Fig. 23) and the lip middle lobe is subrhombic, widest near the middle. Based on the distinct petiole and leaf shape observed in this species it seems to be related to P. modestus, but the middle lobe of P. mollis is longer than wide (vs about equally long and wide in P. modestus).

Figure 23 Psilochilus panamensis.

(A) Dorsal sepal. (B) Petal. C. Lateral sepal. (D–F) Lip. Scale bar = 5 mm. (A–D) Drawn from Folsom 2954 (MO), (E) Drawn from Folsom 6115 (MO), (F) Drawn from Knapp 5778 (MO). Drawn by N. Olędrzyńska.

Additional specimen examined: ECUADOR. Zamora-Chinchipe: At the north slope of the Cordillera del Condor near Paquisha, 1,450 m, 4 February 1987, A. Hirtz 3115, C. Luer, J. Luer (MO!).

18. Psilochilus panamensis Kolanowska (2015: 407).

Type: PANAMA. Veraguas: 6.4 km outside of Santa Fé on the road that passes the agriculture school. Headed toward the cordillera, 5 May 1977, J. P. Folsom 2954 (holotype MO!, isotypes: MO!). Figure 23.

Plants up to 45 cm tall. Leaves 4–5, shortly petiolate; blade 5.5–7 × 2.5–4 cm, ovate to elliptic, obtuse to subacute, occasionally with silver stripes; petiole 0.8–1.5 cm long. Inflorescence up to 4.5 cm long, 4–7-flowered. Floral bracts 9–14 mm long. Pedicellate ovary 12–20 mm long. Flowers with green sepals and pale green petals, lip purple. Dorsal sepal 21–23 × 2.8–3.0 mm, oblong-lanceolate, subacute to acute, 5-veined. Lateral sepals 18–20 × 2.5–4 mm, obliquely linear-oblanceolate acute, 3- or 5-veined. Petals 17.5–18 × 3–3.5 mm, elliptic-lanceolate to oblong-elliptic, obtuse, 3- or 5-veined. Lip 15.5–16 mm long, 5–6 mm wide across lateral lobes; claw 4.2–4.6 mm long; lateral lobes extending up to about middle of the middle lobe, obliquely ovate, rounded to subobtuse at the apices; middle lobe 6–6.5 × 3–4.5 mm, ovate to elliptic, subacute; disc ornamented with three or five thickened, central veins. Gynostemium 16–17 mm long, slender, slightly arcuate in the upper part.

Distribution, habitat and ecology: Localities of this species are distributed along Cordillera Central (Fig. 4C). It grows terrestrially in premontane forest at altitudes of 900–1,700 m. Flowering has been recorded in May, June and October. The highest niche overlap was observed between P. panamensis and P. carinatus (D = 0.4867, I = 0.7645), while the score of this statistic for the most similar species, P. physurifolius was much lower (D = 0.0561; I = 0.2177).

Taxonomic notes: This species resembles P. physurifolius but it is easily distinguished from it by the lip form. In P. physurifolius the lip middle lobe is short, about 4–5 mm long, suborbicular (vs middle lobe up to 6.5 mm long, ovate to elliptic). The other Panamanian Psilochilus species with petiolate leaves, P. dressleri Kolan., is characterized by the subsessile leaves, suborbicular, rounded lip middle lobe and large, obliquely ovate-falcate lip lateral lobes that extend up to two-third of the middle lobe.

Additional specimens examined: PANAMA. Coclé: Hills N of El Valle, E slope and ridges leading to Cerro Gaital, 900–1,000 m, 8°40′N, 80°07′W, 27 June 1982, S. Knapp 5778 (MO!), Chiriqui/Bocas del Toro: Cerro Colorado. Along intersection of Bocas Road with main ridge road. 11.8 km from Chami along path headed into Bocas del Toro, 1,400–1,700 m, 24 October 1977, J. P. Folsom 6115 (MO!).

Incertæ sedis

The specimens from the three following collections may represent undescribed species, however the additional material would be required to confirm this assumption:

McPherson 9679 (MO)—this specimen has petiolate, ovate, obtuse leaves in which it resembles P. modestus, but the lip middle lobe is prominent, ovate-elliptic, almost truncate at the apex. It was identified as P. carinatus by Rothacker, but the lip shape of this species is different.

Herrera 1484 (MO)—this specimen has distinctly petiolate, broad leaves, but unlike P. modestus its lip middle lobe is elongated, elliptic and acute at the apex and the obtuse lateral lobes extend to 2/3 of the middle lobe length. According to the Dressler’s note on the sheet this orchidologist considered it as a new species (“possibly unnamed”). It was also identified as P. carinatus by Rothacker, but the lip shape of this species is completely different than in specimen collected by Herrera.

von Türckheim 3134 (BM, NY, W)—this specimen has sessile leaves, but unlike other representatives of P. macrophyllus-complex its lip middle lobe is large, constituting about 1/3 of the total lip length and the lip claw is long, constituting over 1/3 of the lip length.

Morphometry

Morphological variation of the studied taxa visible on the PCA analysis diagrams was relatively large. However, the specimens have not created a clear pattern of grouping, and their taxonomic affiliation was explained only to a small extent (Fig. 24A). This is due to a significant share of characters of minor importance in the performed ordination, as well as to the high morphological similarity between taxa, where their ranges of morphological variation slightly overlapped. The picture of variation has not changed too much when the PCA analysis was performed based on characters with the greatest contributions enhanced in the previous combined analysis (indicated loadings: the lip length and width, as well as the petal, dorsal and lateral sepalslength; Fig. 24B). A somewhat different picture was observed when the analyses were only performed on the basis of the floral characters (Fig. 24C), where the individuals formed smaller groups, consistent with the recognized taxa in most cases (indicated loadings: the lip width, the middle lobe length and the dorsal sepal length).

Figure 24 Principal component analysis (PCA) of Psilochilus taxa.

(A) Based on vegetative and floral characters from selected specimens, (B) For the characters with the greatest contributions, (C) Based on floral characters only.

In turn, the discriminant analysis showed a highly statistically significant differentiating value for the studied taxa with respect to the measured morphological characters (Wilks λ = 0.00; F(71,250) = 2.75; p < 0.0001). Variables describing the lip length (−4.13) and the middle lobe length (6.34), as well as the claw length (−2.50) and the lateral lobes width (2.03) had the largest share in the discrimination of the studied taxa. The cumulative percentage of explained variance was 80%. On the other hand, the cumulative percentage of explained variance increased to 93%, when the discriminant analysis was performed based on the floral characters (Wilks λ = 0.00; F(207,266) = 3.99; p < 0.0001). Variables describing the lip width (1.27) and the petal length (0.94), as well as the lateral lobes length (1.34), the isthmus length (1.30) and the length between claw base and apex of the lip middle lobe (1.46) had also the largest share in the discrimination of the recognized taxa.

Biogeography

Based on the available records of Psilochilus that could be precisely placed on map, the species richness of particular regions within the tropical Americas differs both in number of taxa and species composition. The highest number of Psilochilus species was found in mountainous regions of Colombia (especially Cordillera Occidental and Cordillera Central) and the Middle America, particularly in Panama (Fig. 25). The greatest genus representatives richness area was found in two biogeographical provinces—the North Andean (six species, 50% taxa endemic to this region) and in the Panamanian (five species, 60% endemics) (Fig. 26). Both in Brazilian Planalto and Central American provinces four species can be found but only in the first one, endemic flora was noted (25% of recorded species). According to the level of endemism, the very unique flora was noted also in the Yungas province where one endemic species occurs as well as in the Mardean-Cordilleran where three Psilochilus species were recorded including one endemic. In all other biogeographical provinces (Lesser Antillean, Greater Antillean, Venezuelan Dry Forest, Llanos, Amazonian, Cuban, and Serra do mar) only 1–2 species belonging to this genus were found but with no species restricted to any of these units.

Figure 25 Species richness of Psilochius orchids within the tropical Americas.

The colour gradient indicates an enhanced diversity from zero (white square) to five (black square).

Figure 26 Psilochilus flora in the biogeographical provinces of tropical Americas (divisions follow Udvardy, 1975).

Numbers in the circles indicate the number of Psilochius species for separate regions and the squares give the numbers of taxa common to the provinces shared. P. sanderianus was excluded from the analysis as there is not data about locality of this species.

According to the political borders the highest species diversity is observed in Brazil (six species), Colombia (six species) and Panama (five species). In Ecuador four species were noted, in Costa Rica—three, in Venezuela, Haiti, Jamaica, and Guatemala—two, and in Bolivia, Mexico, Cuba, Dominica, Dominican Republic, Puerto Rico, Martinique, Guyana, Grenada and Peru only one. Three species are restricted in their distribution to Colombia, another three to Panama and two to Brazil. In each Venezuela and Guatemala a single endemic Psilochilus species was recorded till now. All other mentioned countries can be characterized by no endemic species according to the present knowledge.

The Bray-Curtis analysis of similarities among the Psilochilus floras from different regions of the Latin America shows the presence of six main groups (Fig. 27). One of them includes Central and Northern Andes and region of the south-eastern Brazil and covering the Udvardy’s (1975) Northern Andean, Sierra do mer, and Brazilian Planalto biogeographical provinces. The second group composes of the Lesser Antilles, Costa Rica and Panama which are mentioned in biogeographical studies as Panamanian, Central American and Lesser Antillean provinces. The Greater Antilles and Sierra Marde moutains located in Guatemala and southern Mexico or Greater Antillean, Cuban, and Mardean-Cordillieran biogeographical provinces according to Udvardy’s nomenclature made the third group. The second and the third groups made one bigger clade covering all terrestrial areas around the Carribean Sea. The fourth group includes Amazonian and Llanos provinces which are placed in southern part of Venezuela and south-eastern region of Colombia. Venezuelan Dry Forest (northern regions of Venezuela and Colombia) as well as Yungas (south-central Andes) made the last two groups.

Figure 27 Similarities among Psilochilus floras inhabiting regions of the tropical Americas (Bray-Curtis similarity index for presence/absence data).

Ecology, altitudinal distribution and limiting factors

Species of Psilochilus grow usually terrestrially in the forest understory, however two of the examined specimens, which could not be identified to species level, one from Costa Rica, another from Guyana, were reported as epiphytes (Grayum & Jacobs 3722, MO; Clarke et al. 9606, NY). There are also some records of lithophytic specimens (e.g. Hamiton & Davidse 2625, MO).

Plants prefer shady places, however the habitat variation seems to be very high. Psilochilus populations were found in rainforest, low montane humid forest, cloud forest, premontane wet forest, montane wet forest, pine forest, forest with Liquidambar L. (Hamamelidacea) and Pinus ayacahuite C. Ehrenb. ex Schltdl. (Pinaceae), broad-leaved forest with Magnolia hamori R. A. Howard (Magnoliaceae) and Obolinga zanonii Barneby (Fabaceae), bamboo woods, moist tropical mixed hardwood forest with palm and shrub understory as well as in bog.

The variation of the habitat revealed based on information provided on the herbarium specimens labels was confirmed in the niche overlap test (Tables 3 and 4). Generally the climatic niches of the known Psilochilus species are well separated (Table 3). The only exception is P. macrophyllus var. macrophyllus and P. macrophyllus var. brenesii (D = 0.7924, I = 0.9546). It should be noticed that not for all species a well-sampled database was compiled (Pearson et al., 2006).

Table 3 Results of niche overlap test–D statistic.

D	P. macrophyllus var. brenesii	P. carinatus	P. dusenianus	P. macrophyllus var. macrophyllus	P. modestus	P. mollis	P. szlachetkoanus	P. panamensis	P. physurifolius	P. vallecaucanus	
P. macrophyllus var. brenesii	1.0000	0.5583	0.0080	0.7924	0.0088	0.3018	0.1233	0.4406	0.0104	0.0516	
P. carinatus	x	1.0000	0.0131	0.4683	0.0141	0.4474	0.0783	0.4867	0.0155	0.1198	
P. dusenianus	x	x	1.0000	0.0054	0.0217	0.0126	0.0016	0.0078	0.0020	0.0242	
P. macrophyllus var. macrophyllus	x	x	x	1.0000	0.0119	0.3039	0.1297	0.3636	0.0084	0.0528	
P. modestus	x	x	x	x	1.0000	0.0251	0.0259	0.0101	0.0001	0.0771	
P. mollis	x	x	x	x	x	1.0000	0.0562	0.1543	0.0042	0.1229	
P. szlachetkoanus	x	x	x	x	x	x	1.0000	0.0507	0.0135	0.0146	
P. panamensis	x	x	x	x	x	x	x	1.0000	0.0561	0.0797	
P. physurifolius	x	x	x	x	x	x	x	x	1.0000	0.0098	
P. vallecaucanus	x	x	x	x	x	x	x	x	x	1.0000	

Table 4 Results of niche overlap test–I statistic.

I	P. macrophyllus var. brenesii	P. carinatus	P. dusenianus	P. macrophyllus var. macrophyllus	P. modestus	P. mollis	P. szlachetkoanus	P. panamensis	P. physurifolius	P. vallecaucanus	
P. macrophyllus var. brenesii	1.0000	0.8017	0.0532	0.9546	0.0679	0.5932	0.3207	0.7234	0.0755	0.1720	
P. carinatus	x	1.0000	0.0749	0.7242	0.0771	0.7382	0.2019	0.7645	0.0879	0.3371	
P. dusenianus	x	x	1.0000	0.0395	0.0469	0.0555	0.0109	0.0523	0.0062	0.0684	
P. macrophyllus var. macrophyllus	x	x	x	1.0000	0.0825	0.5911	0.3315	0.6341	0.0635	0.1782	
P. modestus	x	x	x	x	1.0000	0.1090	0.0951	0.0555	0.0009	0.1331	
P. mollis	x	x	x	x	x	1.0000	0.1404	0.3598	0.0195	0.3267	
P. szlachetkoanus	x	x	x	x	x	x	1.0000	0.1388	0.0614	0.0376	
P. panamensis	x	x	x	x	x	x	x	1.0000	0.2177	0.2513	
P. physurifolius	x	x	x	x	x	x	x	x	1.0000	0.0269	
P. vallecaucanus	x	x	x	x	x	x	x	x	x	1.0000	

The distribution of almost all species included in ENM analysis is limited by isothermality and temperature seasonality (Table 5). The occurrence of P. dusenianus, P. mollis and P. physurifolius depends also on precipitation and presence of P. vallecaucanus is somewhat correlated with the altitude. The similarities between preferred climatic niches of morphologically similar species are discussed in the “Taxonomic Treatment” chapter.

Table 5 Estimates of relative contributions of the crucial environmental variables to the Maxent models.

Species	P. macrophyllus var. brenesii	P. carinatus	P. dusenianus	P. macrophyllus var. macrophyllus	P. modestus	P. mollis	P. szlachetkoanus	P. panamensis	P. physurifolius	P. vallecaucanus	
Var_1	Bio 3 (31)	Bio 3 (72.8)	Bio 3 (68.1)	Bio 3 (34)	Bio 4 (39.4)	Bio 3 (30.6)	Bio 4 (33.1)	Bio 4 (33)	Bio 3 (37.2)	Bio 3 (76.5)	
Var_2	Bio 4 (29.1)	Bio 4 (7.1)	Bio 18 (12.1)	Bio 4 (29.3)	Bio 3 (17)	Bio 15 (21.6)	Bio 3 (14.8)	Bio 3 (18)	Bio 19 (17.6)	Alt (13.7)	

The most eurytopic species are P. macrophylus and P. carinatus which can be found respectively in four and three recognized vegetation/climatic zones where members of this genus have been recorded (Table 6). Five other species (28%) have been found to occupy two types of habitats, including P. alicjae, P. brenesii, P. physurifolius, P. szlachetkoanus, and P. vallecaucanus. All other taxa (61%) are restricted to only one vegetation zone, including P. crenatifolius, P. dressleri, P. hatschbachi, P. mollis, P. minutifolius, P. panamensis, P. steyermakii, and P. tuerckheimii known from tropical rain forest, P. antioquiensis from tropical moist deciduous forest, and P. maderoi from tropical mountain system.

Table 6 Habitat preferences of Psilochilus.

Macrohabitat classification follows FRA 2000 Report (2001).

Species	Vegetation zone	
Tropical rainforest	Tropical moist deciduous forest	Tropical mountain system	Subtropical humid forest	
P. alicjae			+	+	
P. antioquiensis		+			
P. carinatus	+	+	+		
P. crenatifolius	+				
P. dressleri	+				
P. dusenianus	+	+			
P. hatschbachi	+				
P. macrophylus	+	+	+	+	
P. maderoi			+		
P. modestus	+		+		
P. mollis	+				
P. minutifolius	+				
P. panamensis	+				
P. physurifolius	+	+			
P. steyermakii	+				
P. szlachetkoanus	+	+			
P. tuerckheimii	+				
P. vallecaucanus		+	+		
Total	14	7	6	2	

The present-day data about altitudinal distribution of Psilochulus orchids is very scanty. The occurrence of populations was reported from lowland areas as well as from the mountainous regions. The lowest elevation at which it was found is 3–5 m, in the Brazilian coast (Silva s.n., US). The highest recorded altitude where the genus representative was collected is 2,500–2,700 m (Steyermark 64559, F). Most taxa occur only in the mountainous areas, usually at elevations above 800 m a.s.l. (Fig. 28). This group is composed by 11 species, however only one (P. antioquiensis) is restricted to elevations above 2,000 m. Only four species have been recorded from lowland to mountains, including P. brenesii, P. carinatus, P. dusenianus and P. minutifolius. The largest altitudinal species ranges are characteristic for P. carinatus, P. macrophylus, P. minutifolis and P. panamanensis (all of them with minimum height difference of 1,500 m), while seven species are known as taxa with very restricted altitudinal distribution.

Figure 28 Altitudinal distribution of Psilochilus species in tropical Americas.

Boxes represent 25th–75th percentiles, upper and lower whisker extends minimum and maximum data point, square insides box indicate median. Psilochilus sanderianus was excluded from the analysis as there is not data about locality of this species.

Phenology

Although the actual knowledge on flowering phenology of orchids classified in genus Psilochilus is far from complete and often is based only on single observations, seven “flowering groups” can be distinguished. The first one includes only one species, P. macrophyllus, in which flowering plants have been observed almost all year (Fig. 29). Except that species only P. panamensis is known as orchid with long flowering period (from spring to early autumn). The third group is made by seven species blooming during winter months (P. alicjae, P. carinatus, P. hatschbachi, P. modestus, P. mollis, P. tuerchheimii, and P. vallecaucanus). Next group is composed by P. dressleri, P. minutifolius, P. physurifolius, and P. szlachetkoanus, in which flowers were recorded in summer period (June–August). The spring-summer group includes only P. antioquiensis. The blooming period for P. steyermarkii is a summer-autumn, while only in autumn flowers of P. duserianus were noted. In case of Psilochilus maderoi and P. sanderianus no data about flowering period are known.

Figure 29 Flowering phenology of Psilochilus species.

Question mark means lack of data.

Discussion and Conclusions

Taxonomy and morphological variation of Psilochilus

Available data allow to recognize 18 Psilochilus species but the discovery of new taxa can be expected as more than half of the known orchids classified in this genus have been described during last five years (Kolanowska & Szlachetko, 2012: Kolanowska & Szlachetko, 2013; Kolanowska, 2013a; Kolanowska, 2013b; Kolanowska, 2014a; Kolanowska, 2014b; Kolanowska, 2015; Kolanowska et al., 2015). At least one more species may occur in Ecuador, and possibly also in Panama, as reported by Rothacker (2007). This author together with Jost proposed even a name for this taxon, “Psilochilus ecuadoriensis” (Rothacker, 2007). The lip shape of this orchid as presented by the authors was not observed in any specimen examined in the present study. The long lip claw and prominent lip lateral lobes resembles those observed in Panamanian P. dressleri Kolan., but the middle lobe shape differs between the two species. Another orchid similar to specimen illustrated by Rothacker in Fig. 4.6. of his dissertation is Colombian P. vallecaucanus Kolan. & Szlach., but in this orchid the leaves are sessile or subsessile (vs. petiolate in “P. ecuadoriensis”). So far “P. ecuadoriensis” is known from a single collection deposited in QCA and since Rothacker has never validly published P. ecuadoriensis, and the collection of Jost, 7955 was not examined, it was omitted in this study.

Psilochilus flowers, which unfortunately have been often ignored during identification of herbarium material in the past, are essential diagnostic character and the species recognition. Without knowledge on the lip morphology of the specimen is basically impossible. Leaves, which always were considered as important feature for species delimitation are usually sessile in the upper part of the stem, hereby for identification purposes only middle leaves should be used. The colour of the leaves seems to be variable within populations of the same species, in some specimens the leaf lower surface is purple to red, in some the upper surface is adorned with silver stripes, in other the leaf is uniformly green. Within species complexes the leaf morphology does not allow to discriminate particular species because its variation among the similar species is not higher than within the species. In conclusion, a large morphological variation among Psilochilus species is observed. Unfortunately, the lack of molecular data makes it impossible to determine which part of this variation is a result of a direct influence of external conditions and it may be not hereditary. The environmental influences may affect any stage of plant development. Each genotype has its own genetically determined level of phenotypic plasticity, with certain characteristics being more conserved than others, e.g. the length and width of labellum or spur, in contrast to e.g. the length and width leaves. The recognition of this phenomenon is extremely important from the taxonomic point of view (Stace, 1991; Naczk et al., 2015). On the basis of the conducted morphometric analyses a continuity of morphological characteristics among the majority of studied species, distinguished in the work as separate taxonomic units was found.

Biogeography and diversity

The highest diversity and/or level of endemism of genus representatives was noted in the Andes (Northern Andean and Yungas biogeographical provinces) as well as in Panamanian province. The data from the Andes clearly confirm the importance of this mountain region as a plant biodiversity hot-spot in South America as it was indicated in numerous previous studies (e.g. Gentry, 1982; Mutke, 2011). Myers et al. (2000) estimated that ca. 50% of all plant species recorded from Andes are endemic to this region, this value is very high also in case of Psilochilus orchids. The very high diversity and endemism of plants observed in the Andean mountains was explained as the consequence of exceptionally rapid surface uplift of Andes during late Miocene and early Pliocene (Garzione et al., 2008; Hoorn et al., 2010; Mulch et al., 2010) as such mountain formation is believed to promote diversification of landscape, what in turn is increasing biodiversity (e.g. Hoorn et al., 2013; Hughes & Atchison, 2015). However, the direct reflection of the geological history in the patterns of diversification and endemism of the Andean plant groups has been recently argued (Antonelli et al., 2009; Antonelli & Sanmartín, 2011; Mutke et al., 2014). Unquestionably, the Andes are one of the most important biodiversity hotspots not only for orchids (e.g. Jost, 2004) but also for many other plant groups (e.g. Huges & Eastwood, 2006; Madriñán, Cortés & Richardson, 2013; Lagomarsino et al., 2016) as well as for animals (e.g. Elias et al., 2009; McGuire et al., 2014).

The second important biodiversity hot-spot of Psilochilus orchids is Panamanian biogeographical province. The occurrence of 27% of known species classified in this genus has been confirmed in this region. Also, the number of endemic species is very high as the province is characterized by 60% level of endemism. Our results strongly correspond with earlier studies made upon Orchidaceae of Mesoamerica. According to Ossenbach, Dressler & Pupulin (2007) and Bogarín et al. (2013) about 10% (ca. 2700 taxa) of all orchid species known all around the world are recorded from this area, of which almost 29% are endemic to Panama. The studies on regional diversity published by Barthlott, Lauer & Placke (1996) indicated that the region from eastern Costa Rica, through Panama to western Colombia has the highest plant biodiversity on the globe. In Panama, the extraordinary biodiversity is a consequence of an unusual mosaic of habitat types and an intermingling of species from both Central and South America, including single species of Psilochilus, but also many other Orchidaceae taxa (D’Arcy, 1987; Condit et al., 1996; Ossenbach, Dressler & Pupulin, 2007; Bogarín et al., 2013).

Noteworthy, high similarity in Psilochilus species composition between the Northern Andean province located in the north-western South America, and Sierra do Mer and Brazilian Planalto placed close to Atlantic Ocean in the south-eastern region of this continent was indicated in our analysis. This is a result of disjunct distribution of P. alicjae, P. macrophyllus and P. modestus. As Psilochilus species are plants preferring tropical forest habitats, the actual distribution of these orchids in mentioned area probably can be explained by Landrum’s (1981) hypothesis. The author suggested that during Oligocene mixed forest with tropical and subtropical elements extended across southern South America. As a result of uplift surface of the Andes during the Miocene, this mountain range started to play important role as a natural barrier for humid air masses from the Pacific Ocean. As a consequence of this barrier, in the region located east of the Andes the air humidity significantly decreased, what in turn modified its flora, especially reduced number of tropical plants and promoted much arid species. Although there is no evidence that such scenario is correct for Psilochilus, it was confirmed for many other plant groups, including e.g. Azara Ruiz & Pav., Drimys J. R .Forst. & G. Forst., Mutisia L. f., Perezia (L. f.) Lag., Persea Mill., Alstoemeria L., Araucaria Juss., Myrceugenia O. Berg., Gunnera L., Escallonia Mutis ex L. f., and Schizolobium parahyba (Vell.) S. F. Blake (Landrum, 1981; Margis et al., 2011; Zorzanelli et al., 2016; Murillo-A, Stuessy & Ruiz, 2016).

High similarity of Psilochilus floras between Panamanian, Central American and Lesser Antillean provinces on one side, and between Cuban, Greater Antillean and Mardean-Cordilleran biogeographical units on the other, can be explained by geological history of Central Mesoamerica and the Caribbean. According to James, Lorente & Pindell (2009), at least few important geological events played important role in forming the present-day type of landscape of this region, including e.g. subduction of oceanic crust of the South American Plate under the Caribbean Plate, subduction of the Cocos Plate under the Caribbean Plate as well as volcanic and earthquake activity of the entire area during the last millions years. For example, similar origin period of the island arc of the Lesser Antilles and mountains located in Panama and Costa Rica and near distances of both these regions to South America probably resulted in relatively high similarity of their flora also in other taxonomical groups (Graham, 2011).

Ecology

The analysis of ecological preferences of Psilochius species shows that most species have relatively narrow habitat specialization. Based on available data almost 90% of actually known taxa occur in only one or two vegetation zone types and their altitudinal distribution is usually restricted (Table 6; Fig. 28). Only P. macrophyllus, the species with widest geographical range within the genus, was found in various habitats. The highest Psilochilus diversity was noted in tropical rain forest and tropical moist deciduous forest. Surprisingly, despite that most of the taxa included in the analysis were found in very similar or the same vegetation type, the ENM analysis indicated that in fact most of the species occupy different climatic niches.

Phylogeny

The only phylogenetic study on Psilochilus was conducted by Rothacker (2007) who, however, has never published his doctoral dissertation in peer-reviewed journal. The analysis performed by this author based exclusively on plastid trnL-F spacer region. Unfortunately the trials of extracting DNA from herbarium specimens were unsuccessful and exclusively genetic material obtained from the fresh leaves was used. Rothacker (2007) recognized only seven species of Psilochilus and he was able to gather material from P. macrophyllus (one sample), P. modestus (one sample), P. mollis (nine samples), P. physurifolius (one sample), putative new species “P. ecuadorensis” (one sample) as well as one unidentified species. Psilochilus dusenianus and P. carinatus were not included in his analysis. In the phylogenetic tree presented by the author P. physurifolius was at the base of the genus, followed by the clade of P. modestus and P. macrophyllus which was sister to a larger clade containing P. mollis and “P. ecuadoriensis.” The relationships within the last clade remained not fully resolved. Undoubtedly, the more extensive sampling and the analysis of additional molecular markers is necessary to reveal the actual phylogeny of Psilochilus, especialy that since studies by Rothacker (2007) eleven new species for the science have been described and classified in this genus (Kolanowska & Szlachetko, 2012: Kolanowska & Szlachetko, 2013; Kolanowska, 2013a; Kolanowska, 2013b; Kolanowska, 2014a; Kolanowska, 2014b; Kolanowska, 2015; Kolanowska et al., 2015).

Limitations of the presented study and the future perspectives

The taxonomical and ecological studies on rare tropical plants characterized by broad general geographical range, such as Psilochilus, are often very problematic. The available material in this kind of research often is very limited. The first main reason is that it is not possible to observe existing populations in their known locations distributed in huge territory (in Psilochilus the study area would extend from southern Mexico to south-eastern Brazil). Moreover, it is also impossible to observe most of them in flowering in the same time, and flowers are necessary for correct species identification. For numerous plant species which are difficult to find during field excursions due to their inconspicuous habit, dull flower colour (like in Psilochilus), long dormant period or preference of hardly accessible habitats, the amount of existing data is even more restricted. In such studies herbarium material is very helpful even if the available data are limited. Moreover, herbarium material may constitute the only source of information on populations which are considered as extinct in the nature as a result of habitat loss.

The primary problem associated with herbarium-based research is the necessity of reconstruction of the natural form of vegetative and generative structures. All tissues are subjected to different degree of shrinking and the impact of this process is difficult to predict (Romero-González et al., 2013; Tomaszewski & Górzkowska, 2016). In case of rare plants, like Psilochilus, which are not easily accessible in the natural habitats and practically absent in the horticultural glasshouses, it is also difficult, or even impossible, to evaluate the impact of these deformations by comparison of fresh and dried material.

The limited material may possibly led to inaccurate conclusions regarding ecological preferences, biogeographical patterns and morphological variation of the studied organisms. The information given on herbarium labels about habitat where specimens where collected are usually rather scare and often very general. To reveal differences in preferred climatic niches of the studied orchids, we decided to use the most objective method—the ENM technique. As in any other statistical analysis, this approach would be affected by unrepresentative data used as an input matrix. It is also obvious that over time, with the new data being available, various species undergo changes in status. Some taxa once thought to be two different species are really “variants” of just one species (e.g. Microchilus campanulatus Ormerod and M. glanduliferus Ormerod). On the contrary, what was considered to be one widely distributed species may turn out to be several distinct species (e.g. Epidendrum nocturnum Jacq.). Clearly, such changes force modifications in biogeographical hypotheses. In this paper, a new status was proposed for P. crenatifolius which apparently falls into morphological variation of P. macrophyllus, but still some differences recapitulated in the previous chapter of this paper allow to distinguish it as a variety of the latter species.

We did not have the opportunity to extract DNA from examined Psilochilus specimens and due to the numerous misidentifications observed in course of studying herbarium specimens we decided to not conduct phylogenetic studies based on data available in GenBank. On the other hand we do believe that the key to identification of Psilochilus species presented in this paper will allow other researchers to correctly recognize genus representatives and to reveal the relationships of these orchids in the future. The molecular research could validate the status of morphologically distinguishable taxa known from single localities as well as to verify classification of specimens which were not assigned by us to any accepted species (incertæ sedis). Such genetic approach was recently applied to estimate the number of species delimited within other orchid genus—Ophrys L. The diversity of its representatives was variously evaluated between morphological taxonomists. While Sundermann (1980) accepted only 16 species and 34 subspecies, Delforge (2005) recognized 252 species of bee orchids. The molecular studies (Devey et al., 2008) revealed that some putative Ophrys species arose through hybridization rather than divergent speciation, indicating that the genus has been substantially over-divided at the species level. On the other hand, the multi-gene barcoding and a combined molecular species delineation approach revealed numerous cryptic species within various organisms (e.g. Hebert et al., 2004; Jörger & Schrödl, 2013; Fourie et al., 2015; Zuccarello et al., 2015).

It is obvious that taxonomic nomenclature should be based on empirical knowledge but the categories of classification are subjective and arbitrary by their nature. Traditionally, the hierarchy of ranks represents relative levels of morphological divergence, however, the alternative bases for classification were proposed when molecular data became more available (de Queiroz & Gauthier, 1990; de Queiroz & Gauthier, 1992). We tried to add objectivity in our recognition of the species by conducting PCA analysis, however, it did not clarified boundaries between representatives of Psilochilus. We believe that this was caused by the insufficient number of data and the same situation would be observed in any other studies on rare plants of high morphological variation. Again, if the significant amount of molecular data would be available for the genus representatives, it potentially would be possible to identify particular species on molecular level. However, the locus under study has to be properly assessed before undertaking any taxonomic identification to ensure that there is no overlap between intraspecific variation and interspecies divergence. This should be optimally, individuals should be genotyped, preferentially from different geographic locations (Pereira, Carneiro & Amorim, 2008).

Supplemental Information

Supplemental Information 1 Annex 1. Complete list of examined Psilochius specimens.

Click here for additional data file.

The Curators and staff of the cited herbaria are thanked for their kind hospitality and assistance during visits of the first author and for making specimens available for loan. We are grateful to Prof. Richard Bateman and the anonymous Reviewer for constructive and thought-provoking comments on the manuscript. We are grateful to Natalia Olędrzyńska and Sławomir Nowak for preparing illustrations. Piotr Jóźwiak it thanked for preparing maps used in biogeographical analysis and Prof. Dariusz L. Szlachetko for the valuable comments on the paper. The latter author would like to express special thanks to Barbara Szmalenberg for the never-ending inspiration and showing the beauty of tropical orchids.

Additional Information and Declarations

Competing Interests

Author Contributions

Data Deposition

New Species Registration

The authors declare that they have no competing interests.

Marta Kolanowska conceived and designed the experiments, performed the experiments, analyzed the data, contributed reagents/materials/analysis tools, wrote the paper, prepared figures and/or tables, reviewed drafts of the paper.

Aleksandra M. Naczk performed the experiments, analyzed the data, contributed reagents/materials/analysis tools, wrote the paper, prepared figures and/or tables.

Radomir Jaskuła performed the experiments, analyzed the data, contributed reagents/materials/analysis tools, wrote the paper, prepared figures and/or tables, reviewed drafts of the paper.

The following information was supplied regarding data availability:

The raw data has been supplied as Supplemental Dataset Files.

The following information was supplied regarding the registration of a newly described species:

New species Psilochilus szlachetkoanus LSID 77158122-1; New combination P. macrophyllus var. crenatifolius LSID 77158123-1; P. macrophyllus var. brenesii LSID 77158124-1.

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
