# Peer review of "Herbarium-based studies on taxonomy, biogeography and ecology of Psilochilus (Orchidaceae)"

_PeerJ, doi:10.7717/peerj.2600_

## Round 0.1 · original submission · Minor Revisions

I have now received two reviews that I feel are supportive and constructive. The manuscript looks at a poorly know genus which, due to the limited amount of material available, makes analyses and decisions difficult. This needs to be acknowledge in the manuscript more explicitly so as to point readers to the need for more research. Further the manuscript requires a careful read to pick up the issues the reviewers have highlighted as well as others.

·

Basic reporting

The following review addresses all three categories listed here, primarily addressing 'Experimental' Design and Validity of the Findings:

Review of manuscript submitted to PeerJ on June 2016 by Kolanowska et al. entitled "Taxonomy, biogeography and ecology of Psilochilus (Orchidaceae)"

Summary
This manuscript presents a typical traditional taxonomic treatment of a poorly-known, comparatively challenging genus of tropical plant species, together with some broadly appropriate statistical analyses designed to extract greater interpretation from the information derived from the available specimens. It would be churlish to criticise either aspect of the manuscript; rather, it is the paucity of data available to link the two aspects of the paper that is its great weakness. On balance, I do not believe that this weakness should lead to outright rejection of this manuscript, but I do believe that the authors should be more explicit about that weakness wherever that is appropriate, in both the text and Figures. Necessary changes include noting the number of observations per species in the relevant Figures. In particular, I believe strongly that the present title should be prefixed with the phrase 'Herbarium-based …", so that the reader is immediately aware of the severe limitations of the study.

Review
Before beginning my detailed review, I think I should make clear that I have been repeatedly consulted by PeerJ's editors regarding the basic principles of whether taxonomic work is appropriate for publication in PeerJ. This was primarily because PeerJ's acceptance criteria include requirements that (1) the research question must be relevant and meaningful (it is arguable whether descriptive taxonomy asks any explicit questions), (2) the methods applied must be reproducible (it is questionable whether basic herbarium taxonomy employs explicit methods), and (3) the resulting data must be robust and statistically sound (most descriptive taxonomy is presented qualitatively rather than quantitatively). After much discussion we eventually agreed that, in principle, monographic treatments comparing multiple taxa in a rigorous manner are acceptable for submission to PeerJ.

I outline here this background in order to explain why I previously agreed that this manuscript merited review; the inclusion of mathematical approaches to morphometric analyses, ecological niche modelling, species richness and distribution patterns qualifies this manuscript for review as a potentially scientific work. This recognition leaves as the key question that of whether the materials presented (exclusively a highly limited number of herbarium specimens) are sufficient to justify the use of these statistical methods. If one uses as a yardstick typical herbarium-based publications the answer is definitely Yes. But if one approaches this question from the opposite direction, asking whether the results are statistically or biologically sound, the answer in my opinion is a definite No; this manuscript suffers from three severe weaknesses, one of which is ubiquitous in the discipline of herbarium taxonomy and the other two nearly so.

Firstly, by definition, herbaria take living plants and concert them into de facto fossils. A field botanist seeking orchids would most likely first be attracted by the colour and scent of the flowers, which can readily be quantified in a living plant – but not in herbarium specimens. Epidermal structures and textures are altered or even lost during drying and compression. Organs of herbarium specimens shrink, often both substantially and disproportionately so that their shape also changes (e.g. Parnell et al., 2013, Taxon 62: 1259>). The range of characters that can be quantified is greatly diminished. This loss of information is often exacerbated, as in this case, by a decision my herbarium-based morphometricians to confine their analysis to continuous metric characters, voluntarily omitting meristic, scalar and presence/absence characters. Thus, the present matrix (not given) of 25 characters entirely reflects sizes of structures. In addition to the problem of shrinkage, there are always strong positive correlations between such characters. For example, the lengths of the tepals (labellum, lateral petals, dorsal sepal, lateral sepals) are in most taxa very similar because they are constrained by the size of the bud prior to its opening). This is a particular problem with orchids because even flower size varies considerably according to whether the plant in question is young/unhappy (=small) or old/happy (=large). Add to this the fact that herbarium specimens are rarely representative of the population from which they were taken (e.g. they tend to be larger than the average in the case of herbaceous species), and you inevitably preserve a highly biased dataset.

Secondly (as is also generally the case in studies of fossils), there are usually far too few specimens per species present in herbaria to draw meaningful conclusions regarding species circumscription, biogeographic distribution, or especially ecology and niche preference (these subjects necessitate extensive field study). As this statement is true of almost all herbarium-based taxonomic studies (indeed, most herbarium taxonomists wholly ignore the potentially valuable mathematical methods applied here), I do not feel that I can recommend rejection of this manuscript on this basis, even though this study has in my view yielded at least one order of magnitude insufficient data to make reliable interpretations. The present study is based entirely on just 57 specimens that together are judged to constitute 15 species and two further varieties. One species, Psilochilus macrophyllus, provided 17 of those specimens. If one omits from the calculation that species, plus a few specimens ultimately left incertae sedis, that leaves an average of 2.3 specimens per species. Consequently, concluding statements such as "Almost 90% of actually known taxa occur in only one or two vegetation zone types" (lines 1076-7) become more a study in finger-based arithmetic rather than science when you realise that 2.3 specimens are, by definition, unable to represent more than two vegetation types! Nor is it surprising that a considerable proportion of geographical endemism is inferred here; by definition, a single specimen cannot be anything other than endemic.

Paucity of specimens leads to the third weakness of most herbarium-based studies, which is that reciprocal illumination – essential to circumscribe biologically meaningful taxa – is not in practice possible. In order to be philosophically justifiable, circumscription needs to operate through circular reasoning: a hypothesis of the existence of taxa is first made explicit through naming and type specification, then data are collected to test that hypothesis of circumscription, which is then modified on the light of those data. Here, the 15 species have been established a priori through standard taxonomic practice (i.e. authoritarian judgement!), morphometric data have then been gathered (often supported by summary notes on habitat, locality, altitude), but no evidence is presented that there has been any reciprocal illumination – feedback from the ordinations to the original taxonomy in order to modify it. This reappraisal is actually where pre-science ends and science begins. So in the absence of this rationale, what was the actual purpose of the morphometric work? The impression of reliance on authoritarianism is enhanced by the fact that the present classification is given before the interpretation of those data that have been accumulated. And although in theory PCA is a useful tool facilitating reciprocal illumination, this statement does not apply to DCA, which is often used taxonomically for identifying the most diagnostic morphological characters of taxa but actually requires a priori assignment of plants (or specimens) to taxa, thereby precluding reciprocal illumination. I would have liked the authors to state which of the measured characters contributed most to the PCA axes.

But the present authors evidently do not wish to interpret their PCA plots literally. They state that in the PCA analysis using all morphometric characters "taxonomic affiliation was explained only to a small extent" (line 862). Is not an alternative interpretation that the a priori taxonomy explains the data only to a small extent, and might actually be horribly wrong?! This is perhaps understandable for Figs. 24A and 24B, where literal interpretation would group two specimens of P. alicjae and one of P. macrophylla var. brenesii as one species and the rest of the specimens as a second! Even more worryingly, the three specimens of P. alicjae span almost the entire plot in Figs. 24A and B. The ordination of flower data only (Fig. 24C) is more persuasive, but to my eye it suggests grouping into three species rather than 15. In summary, the ordinations appear to feature as trimmings rather than essential components of an exercise in reciprocal illumination. Admittedly, many more individuals would be needed to make credible the results of any such positive feedback loop.

As is almost always the case in descriptive taxonomic works, the authors make no attempt to define the ranks of species, subspecies (not used in the present study) or variety (used in this study). It would be helpful to know what criteria the authors have applied to distinguish these ranks in this case, not least because in Fig. 24 the three 'varieties' of P. macrophylla show similar levels of phenetic difference to that shown on average by the supposed full species.

Changing topic, it seems to me strange that a solely morphological, specimen-based monograph such as this would be completed a decade after the production of a doctoral monograph (Rothacker, 2007), cited by the present authors, that focused on the same orchid genus (and presumably many of the same specimens) but presented in addition molecular data. Rothacker analysed fewer putative species than the present authors but had the benefit of generating molecular data (specifically, the plastid region trnL-F) from some taxa and thereby establishing a phylogenetic framework. He presented a formal taxonomic account of the genus similar to that in the present study (e.g. Rothacker's account of P. mollis begins "Habit up to 59 cm tall", whereas the description in the present manuscript begins "Plant up to 60 cm tall"!), as well as discussing the biogeographic implications also addressed in the present manuscript. He provided and even better-supported molecular phylogenetic and morphological framework for the genus. Thus, to simply state in the present Introduction that Rothacker made an important contribution" (but not state how) and that many of his specimens lacked flowers appears to me decidedly inadequate.

In particular, the present authors appear to have discerned no value in Rothacker's molecular tree, yet in truth it would surely have provided a useful classificatory framework for their own taxonomic revision. To cite one example, the present authors chide Rothacker for "never validly publishing P. ecuadorensis" (lines 992-3), yet even a cursory glance at his molecular tree would have shown them that this supposed species is actually deeply embedded phylogenetically within P. mollis, effectively precluding its recognition as a separate species as it would render P. mollis paraphyletic. Today, an ITS sequence can be obtained for less than €10 per species, so I find it surprising that the authors did not generate molecular data; ITS sequences would have given them a stronger indicator of species distinctions than has the morphometric analysis (though of course the greatest interpretational strength is gained when both morphological and molecular data are generated from the same individual plants).

More specific issues regarding content

You will note that I have not commented on the content of the formal taxonomic descriptions or the diagnostic keys; their presentation looks acceptable to me, and their content can only be legitimately assessed by someone who is already familiar with the genus Psilochilus.

The latitude x longitude grid that is the basis of Fig. 25 seems too coarse to me to be of much interpretational value (a single mapping unit covers an area larger than that of French Guiana!). Though I admit that it is perhaps appropriate in scale to accommodate the small number of records that are being mapped.

Presentation

As with almost every manuscript ever submitted to a journal, this manuscript was submitted at least one version too soon. The English usage is reasonably good, but it does break down on average two or three time per page. Punctuation is suboptimal, and there is a scattering of typos throughout.

Attention to journal format is mixed; for example, the keywords would be better placed in alphabetical order, and the citations and bibliography require further attention.

Figures 1, 2 and 4 would have been better presented in colour (there is, of course, no charge for colour in PeerJ).

I don't understand why the labellum sketches in Figure 3 are presented before rather than after Figure 4. These labellum sketches in Fig. 3 and Figs. 5–23 could usefully have been presented in a smaller number of Figures, though given that PeerJ is not page-restricted, I suppose it doe not matter greatly.

The labels and key for Figure 24 are given in far too small a font. Also, one symbol – a vertically oriented rectangle – present in the figure is not represented in the key.

Figure 27 would be better inverted, such that 100% similarity and the regional names are placed at the top of the Figure rather than the bottom.

Figures 28 and 29 should have the number of observations per species added to indicate why the patterns evident are weak and unreliable. In addition, Fig. 29 should have the genus name removed (species epithets are sufficient) and the two lines of question marks could each be reduced to a single question mark.

All five Tables are both useful and acceptably presented. But PeerJ regulations require the morphometric data underlying the PCA plots to be included as an additional Figure.

Experimental design

See above review

Validity of the findings

See above review

Additional comments

See above review

Reviewer 2 ·

Basic reporting

Kolanowska and co-authors present the results of a thorough revision, including data on biogeography and ecology, on the currently accepted 18 species of the New World orchid genus Psilochilus. This solid alpha-taxonomic article represents an important contribution to the knowledge of this small rather uncommon orchid genus. After introducing the genus’ taxonomic and scientific history they formulate clear aims (though hypothesis as this work is more descriptive) for the presented research. All the methods and tests applied are thorough and solidly applied.

Experimental design

Their research relies solely on the study of over 170 herbarium and spirit preserved specimens. The key organs studied and therefore the plant parts with the most diagnostic features are according to the authors in Psilochilus the orchid lip and the leaves. To illustrate this, the authors provide line drawing of all here accepted species. Earlier studies on the genus had the disadvantage of including incomplete materials – in particular missing floral characters – and therefore represent incomplete contributions for such kind of research. To elaborate on ecological niches of Psilochilus the authors applied Ecological Niche Modelling with a number of bioinformatic tools. As a surprise the authors point out that most species live in different climatic niches although they are found in the similar or identical vegetation types. Too summarize information on biogeography, species diversity and distribution the authors apply a number of established methods and softwares (e.g. PRIMER 6). All the findings are summarised in illustrative figures and tables.

Validity of the findings

However, there are some caveats. I found it difficult to understand in which subfamily Psilochilus is placed. In fact it is not mentioned at all (should Epidendroideae), but less informed botanists might house it in the Vanilloideae because Vanilleae are mentioned at a certain place. Also it should be pointed out on which grounds the Psilochilus is not related to vanilloids anymore. Clearly, the lack of a molecular phylogeny is a disadvantage for a more thorough evolutionary interpretation of the genus. However, there is a small phylogeny available in the thesis of Rothacker (2007) and this clearly should be mentioned by latest in the discussion. Though this phylogeny is small, it might help to interpret some of the data produced here. Maybe the authors of the here presented manuscript should try in a further step to isolated DNAs from well preserved herbarium vouchers of Psilochilus to receive molecular phylogenetic information. In my experience it is well possible to extract useable DNAs from such materials and definitely worth a try. However, the last few sentences (on herbarium DNA) were not a critic on the here presented work; it is just an idea how to proceed in future.

Additional comments

I think the manuscript should be presented to a native English speaking person to improve the language. There are several small mistakes, which make it sometimes difficult to follow the test. To illustrate the language issue I add some examples:
Line 87: ... of the genus s currently .... ; should be probably: ... is currently ....
Line 106: The important contribution..; should be probably: The most important...
Line 111: ... characteristics of each genus representative,...; should be probably: characteristics of each SPECIES representative

Question: line 122: you mention the materials from the various herbaria are subjected to ‚standard procedures’. Which are those? You should elaborate on those.

---

## Round 0.2 · Minor Revisions

I agree with the reviewers that the manuscript has been improved but that "Herbarium-based" needs to appear in the title and more consideration needs to be given to the molecular work that has been conducted to date at the very least. I would ask if you could also go over the first set of reviews and consider if any further changes can be made to the manuscript.

·

Basic reporting

See General Comments

Experimental design

See General Comments

Validity of the findings

See General Comments

Additional comments

Having been asked to re-review this manuscript, I believe that I should begin by pointing out that this is the third time that I have been sent this text by PeerJ. I do not have the time or the inclination to re-review it in detail, given the rigour of my original review.

Neither the authors nor the receiving editor would expect me to agree with their rebuttals to the more substantial of my original comments. For example, I do not see how presenting to conservationists 'species' that probably have no biological reality actually helps their cause. And the lack of molecular equipment in the author's research facility does not in any way preclude them from collaborating with one of the myriad labs that actually possess them. Given technological and conceptual advances of recent decades, I see little value in continuing to produce taxonomic monographs that lack molecular or autecological data.

But I will adhere to my philosophy that reviewers of manuscripts should challenge authors but should not dictate to them. The one exception here to my own self-imposed rule is that I genuinely believe that the PeerJ editors should insist on the title being given the prefix "Herbarium-based".The authors argue that this fact is evident from the Abstract, but still the revised Background section merely states that "A taxonomic revision of this Neotropical endemic based on morphological data is presented." The term "morphological data" could include more scientific approaches neglected by the authors, such as field-based morphometric studies performed at the population level, or even simply a morphological cladistic analysis.

Setting aside the title, the authors have made genuine efforts to improve the presentation of the manuscript. In my opinion, if they are indeed happy to stand by the present content of the manuscript, they should be allowed to publish it forthwith.

Reviewer 2 ·

Basic reporting

No comments as the same as in the first round.

Experimental design

No comments as the same as in the first round.

Validity of the findings

No comments as the same as in the first round.

Additional comments

Kolanowska and co-authors present the revised manuscript on their article with the title ‘Taxonomy, biogeography and ecology of Psilochilus (Orchidaceae)‘. Changes have been conducting mostly according to the suggestions of the two reviewers. They have added an approximate two page long text on the limitations of their study and future projects. This text is important and a good justification for their approach. However, as in my last comment, I find it a major weakness in the presented manuscript Rothacker’s phylogenetic work is neglected. I can understand that they do not want spoil Rothacker’s as yet unpublished PhD thesis, but I think the phylogeny must be mentioned in the one or the other way (you could say that it is the as yet only phylogenetic proposal within the genus without going more into detail; but whether there were or were not legal issues is in my view not an argument to not cite the tree).

As long as the authors include Rothacker’s work more extended, I have no objections about publishing this work in PeerJ.

In the new text (‘Limitations of the presented study and the future perspectives’) there are some typos, please check it carefully.

---

## Round 0.3 · accepted · Accept

Please carefully read over your manuscript when you receive the proofs as there a number of grammatical and spelling errors.